# A Deep Survey of Fish Health for the Recognition of Useful Biomarkers to Monitor Water Pollution

Graziella Orso, Roberta Imperatore, Elena Coccia, Gianluca Rinaldi, Domenico Cicchella and Marina Paolucci *

Department of Science and Technologies, University of Sannio, Via De Sanctis, snc, 82100 Benevento, Italy; graorso@unisannio.it (G.O.); rimperatore@unisannio.it (R.I.); elecoccia@unisannio.it (E.C.); gianlucarinaldi45@gmail.com (G.R.); domenico.cicchella@unisannio.it (D.C.)
* Correspondence: paolucci@unisannio.it

**Abstract:** The aim of the present study was to evaluate the wild freshwater fish health status using a vast array of biomarkers as predictive factors of pollutant exposure. The European eel (*Anguilla anguilla*) and brown trout (*Salmo trutta fario*), resident in rivers with different degrees of pollution in the South of Italy (Picentino River with good environmental quality and Tusciano River with low environmental quality), were examined using biometric parameters, histopathological and immunohistochemical biomarkers to evaluate the health status and a possible correlation with the water quality. Several alterations identified in the liver positively correlated with water and soil pollutants: hemorrhage ($p \leq 0.05$), cytoplasmic vacuolization ($p \leq 0.01$), hemosiderosis ($p \leq 0.05$), irregular arrangement of hepatocytes ($p \leq 0.01$), lipid accumulation ($p \leq 0.05$), necrosis ($p \leq 0.01$), cellular hyperplasia ($p \leq 0.05$), leukocyte infiltration ($p \leq 0.01$) and melanomacrophages centers (MMC) ($p \leq 0.01$). In the spleen, only hemosiderosis correlated with water and soil pollutants ($p \leq 0.05$). The inflammatory biomarker tumor necrosis factor $\alpha$ (TNF$\alpha$) and ciclooxigenase 2 (COX2) responded to the environmental pollution, as well as the oxidative stress biomarkers superoxide dismutase (SOD2) and 8-Hydroxy-2′-deoxyguanosine (8-OHdG). Erythrocytic nuclear abnormalities and erythrocytic cellular abnormalities were found to be significantly higher in the blood of both the European eel ($p < 0.0001$) and brown trout ($p < 0.001$) in the Tusciano River compared with the Picentino River. Taken together, these results outline the need to increase the number of suitable biomarkers to assess fish health and reinforce the importance of employing additional biomarkers in biomonitoring programs that can be applied to evaluate water quality and in environmental assessment around the world.

**Keywords:** European Anguilla; brown trout; river pollution; histopathological biomarkers; liver; blood; spleen

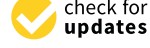



## 1. Introduction

The pollution of the aquatic environment is a serious and growing problem caused by anthropogenic activities, such as industry, agriculture, and trade [1–3]. The resulting low quality of water negatively affects the life of aquatic organisms by inducing stress and compromising the general health status [4]. Rivers and lakes represent the aquatic ecosystems most threatened by the casual release of xenobiotic substances and different kinds of waste [5]. Indeed, the presence of toxic and chemical substances or untreated waste in the water alters the abiotic parameters (pH and dissolved oxygen) of river bodies, negatively affecting the survival of resident organisms [6]. Furthermore, since mollusks, crustaceans and fish species are commonly consumed by humans, the accumulation of pollutants in aquatic organisms could be a risk to human health [7]. Among aquatic organisms, fish are considered the sentinels of water pollution since their health status reflects the quality of the aquatic environment in which they live [8]. By carrying out the

entire biological cycle in water, having a sufficiently long life and moving within more or less extensive areas, fish are often used in biomonitoring programs to assess environmental status [9]. Among fishes, the Teleosts European eel (*Anguilla anguilla*) and brown trout (*Salmo trutta fario*) represent excellent species for the biomonitoring of aquatic ecosystems due to the high degree of diffusion and some characteristics, such as mobility, the capacity to adapt to different salinity conditions and the sensibility to changes in the chemical state of the habitat [9–11].

In fish, exposure to pollutants determines changes at molecular, cellular and tissue levels [8,12]. Subsequently, changes in behavior or external appearance take place [8]. Histological and immunohistochemical analyses, therefore, represent a sensitive and fundamental tool for the early determination of the negative effects induced by toxic and/or polluting substances at the morphophysiological and biochemical level of the organs most sensitive to aquatic pollution, such as gills, livers, spleens and kidneys [13,14]. Performing essential functions such as metabolism and detoxification of toxic substances, the liver and spleen are frequently used as tools for fish health assessment [15]. The identification of biomarkers of these organs may provide a fairly rapid method for detecting exposure to environmental stressors and analyzing the effects of contaminants [16,17]. Several studies have suggested that exposure to contaminants and stressors induces changes in normal hepatocyte organization and cytoplasmic vacuolation, proliferation of white pulp, excessive production of connective tissue (fibrosis) and an increase in melanomacrophage centers (MMCs) [18]. MMCs are crucial to the fish immune system and play a key role in the inflammatory defense [19,20]. Moreover, exposure of fish to several contaminants can also induce the upregulation of some pro-inflammatory factors [1,21–23], including the pleiotropic pro-inflammatory cytokine TNFα (Tumor Necrosis Factor α) and the inflammation marker COX2 (Cycloxygenase 2). In aquatic animals, pesticides and heavy metals, widespread environmental pollutants, could also induce oxidative stress and compromise the antioxidant defense system, generating consequent alterations to vital biomolecules, such as lipids, protein and DNA [5].

Mediterranean rivers are among the most threatened ecosystems due to anthropogenic insults [24]. The continuous polluting threats determine the degradation of the ecological status of aquatic habitats, which is the basis of the worrying decline in fish populations [25,26]. Thus, it is very important to identify in due time the direct or indirect effects of pollution caused by human-induced activities on living organisms in order to take the necessary counteractive measures. In this context, the aim of the present study was to evaluate the wild freshwater fish health status using a vast array of health biomarkers employed as predictive factors of pollutant exposure. For this purpose, the European eel and brown trout, resident in rivers with different degrees of pollution (Picentino River with good environmental quality and Tusciano River with low environmental quality), were examined using biometric parameters and histopathological and immunohistochemical analyses to evaluate the health status and possible adaptations of fish.

## 2. Materials and Methods

### 2.1. Sampling Sites

In this study, two different rivers located in the Campania region (South of Italy) were selected based on different levels of water and soil contamination (Figure 1).

The Picentino River (site of sampling: 40°42′27.5″ N and 14°56′39.6″ E) originates from the Picentini Mountains, and the total area of the hydrographic basin is 150 km$^2$. The Tusciano River (site of sampling: 40°35′58.9″ N and 14°56′57.4″ E) also originates from the Picentini Mountains, and along its course, it crosses upstream a territory covered with wooded vegetation, which towards the valley gives way to intensely urbanized and industrialized areas, suffering all the effects of the anthropic environmental alterations. The Picentino river basin is most affected by the presence of agricultural activities, while in the Tusciano river basin, intensive agricultural activities are complemented by industrial activities.

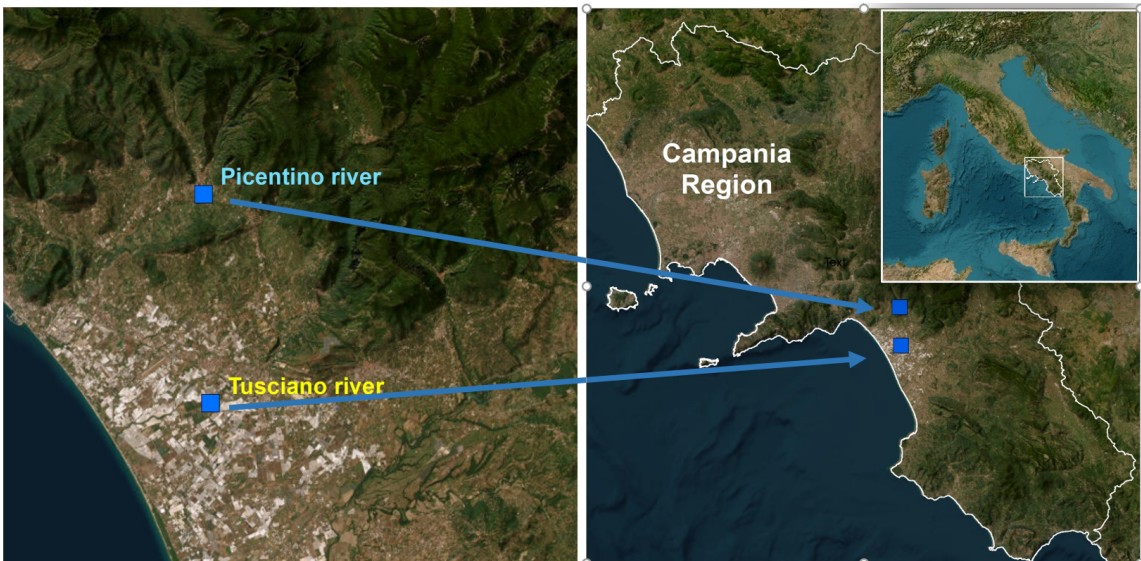

**Figure 1.** Geographical location of the sampling sites.

In order to establish the environmental health status of the two rivers, concentration data of potentially toxic elements (PTEs) and persistent organic pollutants (POPs) in river water and soil near the fish sampling sites were used. These data are part of a very large and detailed environmental monitoring program of the entire Campania Region carried out in 2018/2019 (they are freely available at the link http://www.campaniatrasparente.it/gis.php; accessed on 16 November 2023). Specifically, the data refer to the concentration in water and soils of potentially toxic elements (PTEs) (Sb, As, Be, Cd, Cr, Co, Cu, Pb, Hg, Ni, Se, Tl, Sn, V, Zn), polycyclic aromatic hydrocarbons (PAHs), polychlorinated biphenyls (PCBs) and organochlorine pesticides (OCPs)

### 2.2. Animals and Biological Indices

Adult European eel (*Anguilla anguilla*) (*n* = 10; 5/site) and brown trout (*Salmo trutta fario*) (*n* = 10; 5/site) were collected during the winter season (period November–December 2020) using electroshock equipment. Blood was promptly withdrawn from the caudal tail vessels using a heparin-rinsed syringe with a 23-gauge needle. Fish were kept refrigerated and transported to the laboratory. The total length (TL) was measured with a ruler. The body weight (BW) was measured with a scale to the second decimal place. The liver and spleen were removed and weighed. Physiological integrative indicators such as condition factor (K) and hepatosomatic index (HSI) were calculated. The condition factor (K) was calculated as $K = 100 * BW/TL^3$. The HSI was calculated as the ratio of liver weight to body weight.

### 2.3. Histopathological Analysis

The livers and spleens were fixed in 4% formalin in phosphate buffer pH 7.4, processed according to Coccia et al. [27] and embedded in paraffin. Then, 5-µm sections were stained with hematoxylin-eosin (H&E) and Gomori trichrome. The histological sections were examined under an optical microscope with a Leica DMI6000 equipped with a Leica DFC340 CCD digital camera (Leica Microsystems), and microscopic images were acquired at 10×, 20× and 40× objective magnifications.

Identification of histological alterations was primarily based on the work of Mumford et al. [28] and Martins et al. [29]. The evaluation of histological alterations was carried out by examining 10 sections and 20 fields per section of each organ, with each section selected at a 50 µm distance. The indices of the histopathological alteration (HA) of the livers and spleens were calculated by adapting the scoring system suggested by Bernet

et al. [12]. Briefly, according to this semiquantitative method, each observed alteration was assigned an importance factor (ω) ranging from 1 to 3 (Table 1). Specifically, importance factor 1 corresponded to minimal and reversible damages (circulatory, cellular, and structural alterations), factor 2 corresponded to moderate and reversible damages (inflammation and lipid accumulation), and factor 3 corresponded to irreversible alterations that determine the loss of organ function (cellular death).

**Table 1.** Description of tissue alterations and their importance factor (ω).

| General Description | Alterations | Importance Factor (ω) |
|---|---|---|
| Alterations result from a pathological condition of blood and tissue fluid flow | Blood sinusoid dilation | 1 |
| | Hemorrhage | 1 |
| Alterations result in functional redution or loss of the organ. | Cytoplasmic vacuolization | 1 |
| | Hemosideriosis | 1 |
| | Irregular arrangement of hepatocytes | 1 |
| | Lipid accumulation | 2 |
| | Necrosis | 3 |
| | Cellular hyperplasia | 2 |
| Alterations result from increased presence of cells used in tissue repair; response to damaged tissue | Leukocytes infiltration | 2 |
| | Melanomacrophage aggregates | 1 |
| | Free melanomacroph ages | 1 |

Moreover, the degree and extent of the alterations observed in the livers and spleens were evaluated by using a score value (a) ranging from 0 to 3 (0 = no alterations or very mild changes in <5% of fields; 1 = slight and localized modifications in <20% of fields; 2 = moderate and mild diffuse modifications in 20 to 60% of fields; 3 = severe and widespread modification in >60% of fields). This ranking was used by Nataraj et al. [30] to define a general assessment value of the histopathological lesion in animal tissues. Using the factors of importance (ω) and score values (a), the histopathological alteration indices (HA) were calculated using the following equation:

$$HA = \Sigma \, (\omega \times a)$$

To compare the degree of hepatic damage of the European eel and brown trout in the two different rivers, the liver histopathological index (LHP) was calculated for each fish by using the formula:

$$LHP = \Sigma \, HA$$

The LHP value was interpreted according to the organ evaluation indices proposed by Zimmerli et al. [31]. The LHP was divided into four different ranges of values, of which the range of 0–10 indicated normal structural organization of the organ, the range of 11–20 indicated mild structural changes in the organ, the range of 21–40 indicated moderate structural changes, and the range >40 indicated the presence of severe alterations. For the histopathological analysis of the spleen, the number of MMCs and the percentage of the area of fibrosis were evaluate. The number of melanomacrophages centers (MMCs) was calculated using ten randomly selected splenic sections stained with H&E and counted on the entire section. Measurements were made using the image analysis software Motic Image Plus version 3.0. The fibrosis surface areas were quantified in ten randomly selected splenic sections stained with Gomori Trichrome using the "IHC Toolbox" for Image J software (NIH) (https://imagej.net/ij/plugins/ihc-toolbox/; accessed on 16 November 2023) [32]. All the histological analyses were performed by two independent operators blinded to the type of sample in order to make the results more objective.

### 2.4. Immunohistochemical Analysis

For immunohistochemical analysis, the following primary antibodies were used: anti-TNF$\alpha$ (tumor necrosis factor $\alpha$) (Novus, Cat No. 19532; 1:100); anti-COX2 (cicloxygenase 2) (Cayman, No. 160126; 1:200); anti-SOD2 (superoxide dismutase) (Genetex, Cat No. 124294; 1:100); anti-CAT (catalase) (Genetex, Cat No. 124357; 1:100); anti-8OHdG (8-Hydroxy-2′-deoxyguanosine) (Abcam, Cat No. 48508), 1:100). Sections of liver and spleen were hydrated, and 3% hydrogen peroxide ($H_2O_2$) was used to block endogenous peroxidase. After blocking, primary antibodies were applied using their respective dilutions overnight at 4 °C. The secondary antibody incubation was followed by 1 h with the avidin–biotin complex diluted in Tris-buffered saline according to the manufacturer's instructions (ABC Kit; Vectastain, Vector). Thereafter, a 10-min incubation with 0.05% of 30-diaminobenzidine (DAB) (DAB Sigma Fast, Sigma-Aldrich, Milano, Italy) was applied. Leica DMI6000 light microscopy (Leica Microsystems, Wetzlar, Germany) was used to examine the sections. Images were acquired using a digital camera (JCV FC 340FX, Leica) at a magnification of 20×. A control-staining procedure was carried out to confirm the specificity of the immunoreaction. The density of biomarkers-positive immunosignal was quantified with densitometric analysis on each sample divided into six regions ($n = 3$ animals/site; $n = 6$ pairs of sections/animal, each section selected at 50 μm distance). Optical density was performed with the image analysis software Image Pro Plus® 6.0 (MediaCybernetics, Rockville, MD, USA). All the histological analyses were performed by an independent operator blinded to the type of treatment.

### 2.5. Blood Smear and Staining

A drop of fresh blood (20 μL) from each fish was smeared on a glass slide, dried and stained with May Grunwald-Giemsa to assess the morphological changes in erythrocytes [33]. To perform the blood qualitative–quantitative analysis, three smears were analyzed for each sample, and four regions of interest were examined for each smear. The percentage of erythrocytic nuclear abnormalities (ENA: micronucleus, blebbed nucleus, notched nucleus), erythrocytic cellular abnormalities (ECA: erythrocyte deformed, elongated, or circular) and erythrocytic fusion (EF) was analyzed by using Image Pro Plus® 6.0 image analysis software (MediaCybernetics).

### 2.6. Statistical Analyses

All data were analyzed statistically using GraphPad Prism 8 software (GraphPad, Inc., San Diego, CA, USA), and the histopathological data were presented as Mean $\pm$ Standard Deviation (SD), while optical density and ENA, ECA and EF were reported as Mean $\pm$ Standard Error (SE). Comparison between two sampling sites was performed using a *t*-test or two-way ANOVA. *p*-values below 0.05 were considered statistically significant. A correlation analysis was applied to investigate the relationship between environmental pollutants in stream water and soil and histopathological alterations of tissues. The *p*-value is from a two-tailed test with a confidence interval of 95%. Statistical differences were considered significant whenever $p < 0.05$, and statistical output was represented by stars as follows: nonsignificant (ns) > 0.05, * $p \leq 0.05$, ** $p \leq 0.01$.

## 3. Results

### 3.1. Chemical Pollution in the Tusciano and Picentino Rivers

The data relating to the chemical analyses of stream water and soil sampled near riverbanks clearly show a lower environmental quality for the Tusciano River (Tables 2 and 3).

The water of the Picentino River shows normal values of Eh (387 μS/cm), oxygen (9.4 mg/L) and pH (7.2), while the Tusciano River shows quite altered values of these parameters (1680 μS/cm, 7.8 mg/L and 8.2, respectively). This difference is even more evident for nitrates, which in the water of the Tusciano River are present in concentrations of an order of magnitude greater than those in the waters of the Picentino River (39 versus 2.8 mg/L). The concentrations of PTEs are all below the detection limit in the Picentino

River, while in the Tusciano River, Hg and As, in particular, are present in significant concentrations (0.23 µg /L and 3.13 µg /L).

**Table 2.** Water analysis of the Pcentino and Tusciano Rivers.

|  |  | Picentino River | Tusciano River |
|---|---|---|---|
| Physicochemical parameters | Conductivity (µS/cm) | 387 | 1680 |
|  | Oxygen (mg/L) | 9.4 | 7.8 |
|  | pH | 7.2 | 8.2 |
|  | Temperature (°C) | 14.8 | 15.3 |
|  | Total Inorganic Carbon mg/L | 45 | 86 |
|  | Chlorides mg/L | 7.4 | 22 |
|  | Bromine mg/L | <0.1 | 0.2 |
|  | Nitrates mg/L | 2.8 | 39 |
| PTEs | As (µg/L) | <1 | 3.13 |
|  | Cd (µg/L) | <0.1 | 0.17 |
|  | Co (µg/L) | <0.5 | 0.8 |
|  | Hg (µg/L) | <0.1 | 0.23 |
|  | Ni (µg/L) | <0.5 | 0.77 |
|  | Pb (µg/L) | <1 | 1.22 |
| Pesticides | Boscalid (µg/L) | <0.01 | 0.44 |
|  | Glifosate (µg/L) | <0.01 | 0.32 |
|  | AMPA (µg/L) | <0.01 | 0.85 |
|  | Propizamide (µg/L) | <0.01 | 0.02 |
|  | Terbutryn (µg/L) | <0.01 | 0.05 |
| Fungicide | Metalaxil (µg/L) | <0.01 | 0.04 |
| Other contaminants | Tetrachlorethylene (µg/L) | <0.01 | 0.17 |
|  | Toluene (µg/L) | <0.01 | 0.15 |
|  | Trichloroethylene (µg/L) | <0.01 | 0.11 |
|  | Trichloromethane (µg/L) | <0.01 | 0.14 |

**Table 3.** Soil analysis of the Picentino (PI) and Tusciano (TU) Rivers' drainage basins.

| PTEs (mg/kg) | PI | TU | PAHs (mg/kg) | PI | TU | PCBs (µg /kg) | PI | TU | OCPs (µg /kg) | PI | TU |
|---|---|---|---|---|---|---|---|---|---|---|---|
| As | 8.7 | 12.5 | Total PAHs | 0.0155 | 0.0182 | PCB 77 | 0.003 | <0.003 | Hexachlorobenzene | 0.005 | 0.113 |
| Be | 3.3 | 5.8 | Dibenzo(a,i)pyrene | <0.005 | 0.0051 | PCB 118 | <0.003 | 0.032 | $\alpha$-Chlordane | <0.005 | 0.078 |
| Cd | 0.32 | 0.21 |  |  |  | PCB 105 | <0.003 | 0.02 | Chlordane | 0.005 | 0.081 |
| Co | 5.7 | 11.2 |  |  |  | PCB 167 | <0.003 | 0.004 | o,p'-DDE | <0.005 | 0.059 |
| Cu | 38.8 | 35.8 |  |  |  | PCB 156 | 0.004 | 0.007 | p,p'-DDE | 0.02 | 7.591 |
| Cr | 10.4 | 18.3 |  |  |  | PCB 157 | 0.006 | <0.003 | o,p'-DDD | <0.005 | 0.019 |
| Hg (µg /kg) | 23 | 23 |  |  |  | PCB 189 | 0.004 | <0.003 | p,p'-DDD | <0.005 | 0.109 |
| Ni | 10.5 | 12.6 |  |  |  | PCB-28 | <0.003 | 0.005 | o,p'-DDT | <0.005 | 0.292 |
| Pb | 36.1 | 39.9 |  |  |  | PCB-52 | <0.003 | 0.006 | p,p'-DDT | <0.005 | 1.771 |
| Sn | 2.5 | 2.7 |  |  |  | PCB-101 | <0.003 | 0.012 | Dieldrin | <0.005 | 0.191 |
| Tl | 0.8 | 1.36 |  |  |  | PCB-153 | 0.135 | 0.042 | Endosulphan sulphat | 0.128 | <0.005 |
| V | 45 | 80 |  |  |  | PCB-138 | 0.065 | 0.043 | DDD, DDT, DDE | 0.0325 | 9.84 |
| Zn | 77.1 | 75.6 |  |  |  | PCB-180 | 0.063 | 0.055 |  |  |  |
|  |  |  |  |  |  | Trichlorobiphenyls | <0.01 | 0.012 |  |  |  |
|  |  |  |  |  |  | Tetrachlorobiphenyls | <0.01 | 0.036 |  |  |  |
|  |  |  |  |  |  | Pentachlorobiphenyls | <0.01 | 0.121 |  |  |  |
|  |  |  |  |  |  | Hexachlorobiphenyls | <0.01 | 0.16 |  |  |  |
|  |  |  |  |  |  | Heptachlorobiphenyls | 0.227 | 0.098 |  |  |  |
|  |  |  |  |  |  | Octachlorobiphenyls | 0.081 | 0.013 |  |  |  |
|  |  |  |  |  |  | Nonachlorobiphenyls | 0.059 | 0.032 |  |  |  |
|  |  |  |  |  |  | Decachlorobiphenyls | 0.051 | 0.064 |  |  |  |
|  |  |  |  |  |  | Total PCB | 0.418 | 0.536 |  |  |  |

In the stream water of the Tusciano, it should also be noted the massive presence of pesticides, which are completely absent in the Picentino River. However, it should be specified that the concentration values of these pollutants, although significant, are below the thresholds established by Italian Legislative Decree 152/2006 for water.

The same situation is observed in the soil. Here, the concentration of both PHAs and PCBs is higher, but the difference is quite evident in the concentrations of DDD, DDT and DDE, which in the Tusciano River basin reach 9.84 µg/L, very close to the threshold limit

of the legislative decree 152/2006 for soils (10 μg/L), while in the Picentino river basin, the concentration values are much lower (0.0325 μg/L).

### 3.2. Biological Indices

The biological indices of European eel and brown trout from the Picentino and Tusciano Rivers are reported in Table 4. No significant differences in K ($p > 0.05$) were detected. The hepatosomatic index (HSI) was significantly ($p < 0.05$) higher in both species in the Tusciano River compared to the Picentino River.

**Table 4.** Biological indices of European eel (*A. anguilla*) and brown trout (*S. trutta fario*) from the Picentino and Tusciano Rivers. Values are presented as mean ± SD.

| Species | River | BW (g) | TL (cm) | K | HSI (%) |
|---|---|---|---|---|---|
| *European eel* | Picentino | 44.2 ± 4.06 | 29.5 ± 4.09 | 0.18 ± 0.05 | 1.61 ± 0.25 |
| (*A. anguilla*) | Tusciano | 41.8 ± 1.97 | 29.1 ± 2.54 | 0.17 ± 0.04 | 1.98 ± 0.03 [a] |
| *Brown trout* | Picentino | 28.8 ± 4.70 | 13.3 ± 0.73 | 1.22 ± 0.09 | 1.09 ± 0.24 |
| (*S. trutta fario*) | Tusciano | 27.9 ± 4.92 | 13.5 ± 1.07 | 1.16 ± 0.28 | 1.72 ± 0.09 [b] |

Different letters indicate statistical difference between samples within the river; (a: $p < 0.05$; b: $p < 0.001$).

### 3.3. Liver Histopathology

The liver sections of European eel and brown trout from the Picentino River showed a compact structure characterized by homogenous parenchyma, polygonal hepatocytes having a spherical nucleus surrounded by a network of sinusoids (Figure 2A,C). Several alterations in the hepatic tissues of both fish species from the Tusciano River were present (Figure 2B,B$_1$,D,D$_1$).

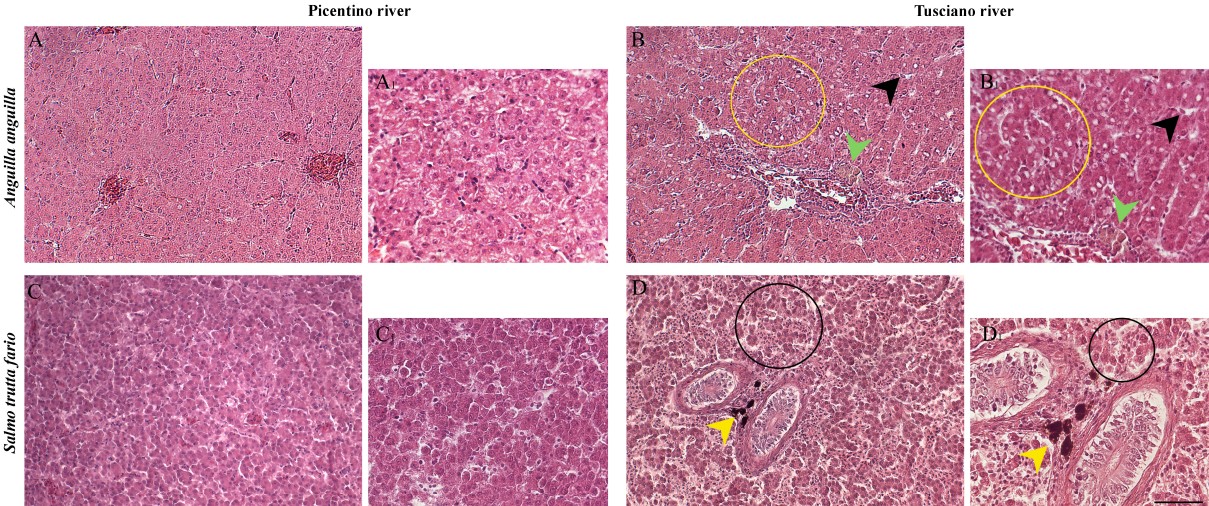

**Figure 2.** Histopathological assessment of the liver of European eel (*A. anguilla*) from the Picentino (**A**,**A1**) and Tusciano Rivers (**B**,**B$_1$**) and brown trout (*S. trutta fario*) from Picentino (**C**,**C$_1$**) and Tusciano rivers (**D**,**D$_1$**) (20× and 40×, H&E). (**A**,**A$_1$**) Liver of European eel from the Picentino River showing normal morphology; (**B**,**B$_1$**) liver of European eel from the Tusciano River showing cytoplasmic vacuolization (yellow circle), sinusoid dilation (black arrow), hemosiderosis (green arrow); (**C**,**C$_1$**) liver of brown trout from the Picentino River showing normal morphology; (**D**,**D$_1$**) liver of brown trout from the Tusciano River showing accumulation of melanomacrofages centers (MMCs) (yellow arrow), irregular arrangement of hepatocytes (black circle). Scale bar: 150 μm, and 75 μm for the higher magnifications.

The main hepatic alterations identified were (Table 5): (1) leukocyte infiltration; (2) necrosis; (3) hemorrhage; (4) irregular arrangement of hepatocytes; (5) dilation of the sinusoids; (6) cytoplasmic vacuolization; (7) cellular hyperplasia; (8) hemosiderosis; (9) congestion of blood vessels; (10) MMCs.

**Table 5.** Indices of histopathological alterations (HA) of the liver of European eel (*A. anguilla*) and brown trout (*S. trutta fario*) from the Picentino and Tusciano Rivers. Scores are presented as mean values ± SD. Liver alterations were scored as follows: 0 = none, 1 = mild, 2 = moderate and 3 = severe.

| Alterations | European Eel (*A. anguilla*) | | Brown Trout (*S. trutta fario*) | |
|---|---|---|---|---|
| | **Picentino River** | **Tusciano River** | **Picentino River** | **Tusciano River** |
| Blood sinusoid dilation | 0.25 ± 0.46 | 1.25 ± 0.70 [b] | 0.37 ± 0.51 | 1.87 ± 0.83 [c] |
| Hemorrhage | 0.5 ± 0.53 | 2 ± 1.07 [b] | 0.63 ± 0.51 | 1.75 ± 1.03 [a] |
| Cytoplasmic vacuolization | 0.37 ± 0.51 | 1.5 ± 0.53 [c] | 0.5 ± 0.53 | 1.62 ± 0.91 [b] |
| Hemosideriosis | 0.5 ± 0.53 | 1.25 ± 0.70 [a] | 0.62 ± 0.51 | 1.37 ± 0.74 [a] |
| Irregular arrangement of hepatocytes | 0.25 ± 0.46 | 1.63 ± 0.74 [c] | 0.37 ± 0.51 | 1.75 ± 0.70 [c] |
| Lipid accumulation | 0.5 ± 0.92 | 1.5 ± 0.92 [a] | 0.25 ± 0.70 | 2.0 ± 1.06 [b] |
| Necrosis | 0.37 ± 1.06 | 3.37 ± 1.92 [b] | 0.75 ± 1.38 | 3.75 ± 1.38 [c] |
| Cellular hyperplasia | 0.25 ± 0.70 | 2.75 ± 2.12 [b] | 0.5 ± 0.92 | 2.25 ± 1.66 [a] |
| Leukocytes infiltration | 0.5 ± 0.92 | 2.5 ± 0.92 [c] | 0.75 ± 1.03 | 2.75 ± 1.84 [a] |
| melanomacrophages centers (MMCs) | 0.62 ± 0.51 | 1.37 ± 0.52 [a] | 0.62 ± 0.74 | 1.25 ± 0.46 |

Different letters indicate statistical difference between samples within the river; (a: $p < 0.05$; b: $p < 0.001$; c: $p < 0.0001$).

The calculation of the LHP index (Figure 3) allowed the comparison between the degree of damage in European eel and brown trout from the Picentino and Tusciano Rivers. Specifically, the specimens sampled in the Picentino River showed a very low LHP index (4.12 ± 2.79 for European eel and 5.37 ± 2.51 for brown trout). In the specimens sampled in the Tusciano River, on the other hand, there the LHP was significantly higher ($p < 0.0001$) in both species, highlighting the presence of moderate tissue damage.

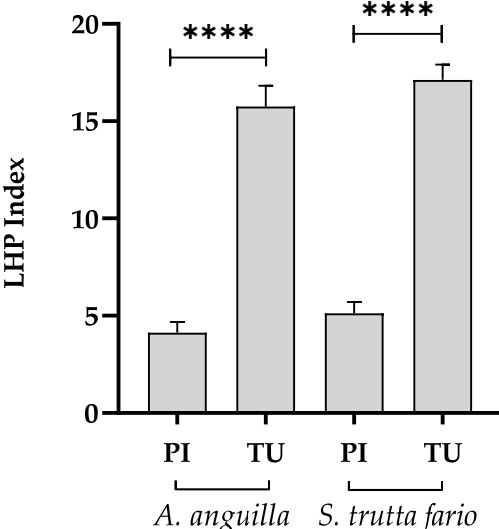

**Figure 3.** Liver histopathological index (LHP) of European eel (*A. anguilla*) and brown trout (*S. trutta fario*) of the Picentino (PI) and Tusciano (TU) Rivers. Values are presented as mean ± SD. * indicates statistical difference between sampled sites (**** $p < 0.0001$).

### *3.4. Liver Immunohistochemistry*

#### 3.4.1. Inflammatory Biomarkers

Figures 4 and 5 illustrate the immunopositivity of TNFα and COX2 in the liver of the European eel and brown trout from the Picentino and Tusciano Rivers. In the liver of European eel, the immunopositivity of TNFα and COX2 appeared weaker in all specimens sampled in both the Picentino (Figures 4A and 5A) and Tusciano Rivers (Figures 4B and 5B). However, the densitometric analysis showed a significant increase in TNFα in the European eel from the Tusciano River compared with the Picentino River ($p < 0.0001$). In the brown trout, a high immunoexpression for both biomarkers of inflammation was observed in the liver of specimens sampled in the Tusciano River (Figures 4D and 5D), while a slight

immunopositivity was detected in the liver of the specimens sampled in the Picentino River (Figures 4C and 5C). Specifically, the brown trout from Tusciano showed a significant increase in both TNFα ($p < 0.05$) and COX2 ($p < 0.05$) immunodensity compared with the Picentino River.

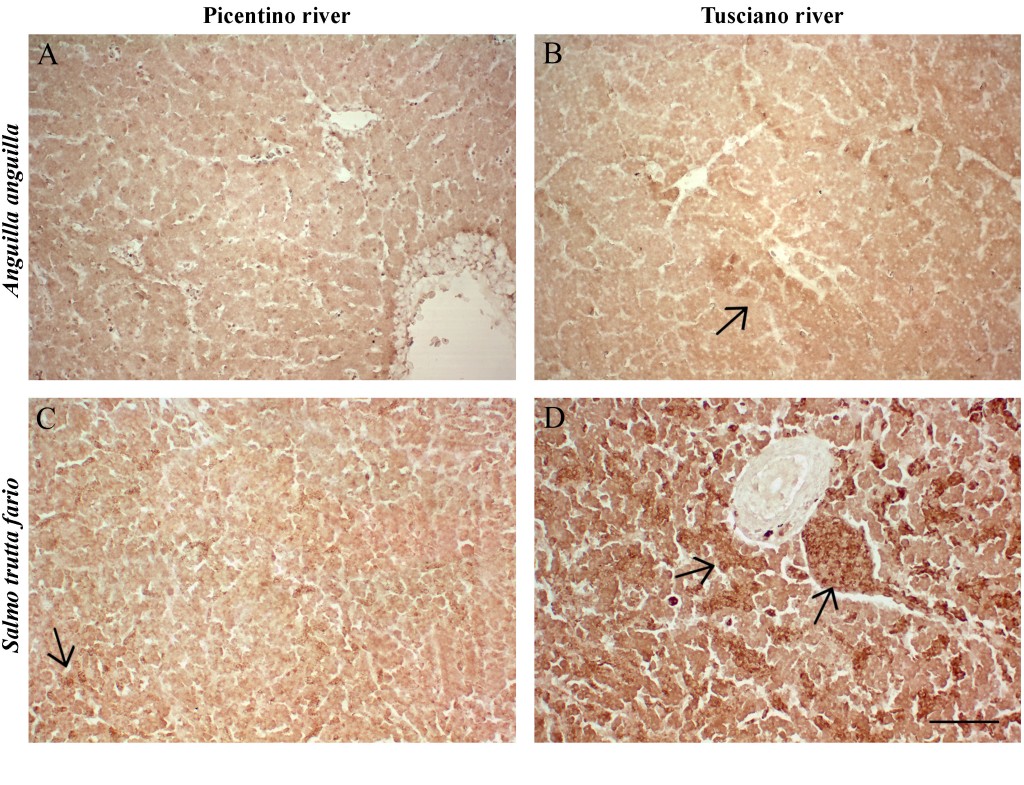

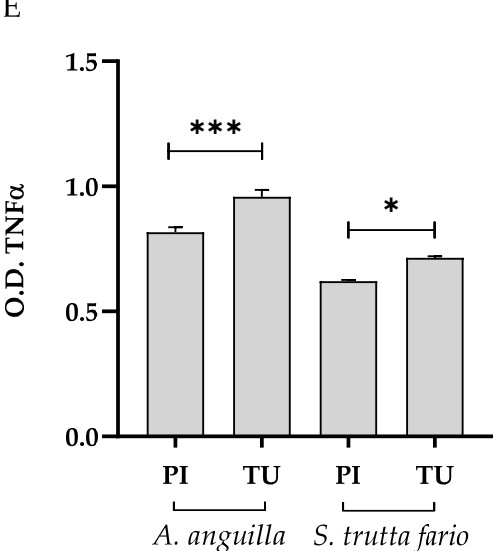

**Figure 4.** Immunohistochemical analysis of Tumor Necrosis Factor α (TNFα, 20×) in the liver of European eel (*A. anguilla*) from the Picentino (**A**) and Tusciano Rivers (**B**) and in the liver of brown trout (*S. trutta fario*) from Picentino (**C**) and Tusciano rivers (**D**). Arrows indicate immunopositivity. Scale bar: 150 μm. (**E**) Bar graph shows TNFα optical density (O. D.) in the liver of European eel (*A. anguilla*) and brown trout (*S. trutta fario*) from the Picentino (PI) and Tusciano (TU) Rivers. Values are presented as mean ± SD. * indicates statistical difference between sampled sites (*** $p < 0.0001$; * $p < 0.05$).

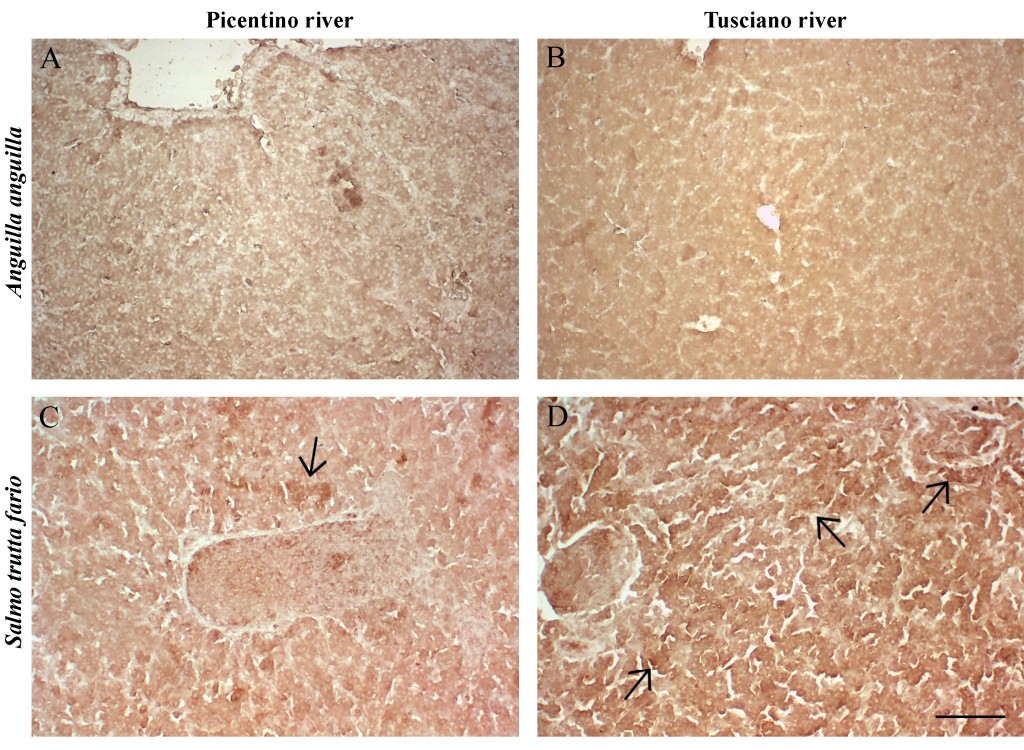

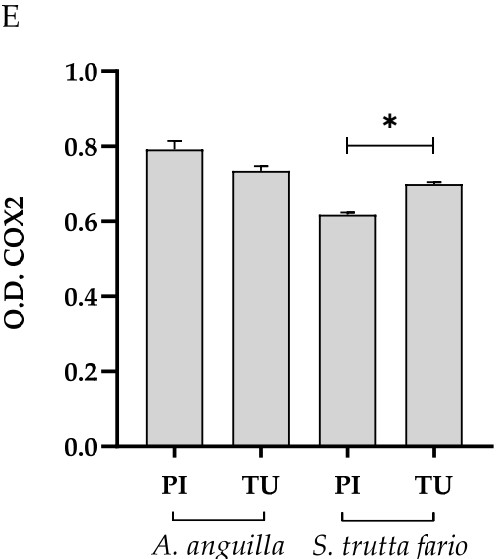

**Figure 5.** Immunohistochemical analysis of Cycloxygenase 2 (COX2, 20×) in the liver of European eel (*A. anguilla*) from the Picentino (**A**) and Tusciano Rivers (**B**) and in the liver of brown trout (*S. trutta fario*) from the Picentino (**C**) and Tusciano Rivers (**D**). Arrows indicate immunopositivity. Scale bar: 150 μm. (**E**) Bar graph shows COX2 optical density (O. D.) in the liver of European eel (*A. anguilla*) and brown trout (*S. trutta fario*) from the Picentino (PI) and Tusciano (TU) Rivers. Values are presented as mean ± SD. * indicates statistical difference between sampled sites (* $p < 0.05$).

### 3.4.2. Antioxidant Biomarkers

Figures 6–8 report the immunopositivity of CAT, SOD2 and 8-OHdG in the liver of European eel and brown trout from the Picentino and Tusciano Rivers. The hepatic sections of European eel and brown trout from the Picentino River show a slight immunoexpression of the antioxidant enzymes CAT (Figure 6A,C) and SOD2 (Figure 7A,C) and an absent immunopositivity for 8-OHdG (Figure 8A,C). Both European eel and brown trout of the

Tusciano River did not show significant differences in CAT immunodensity (Figure 6E), while a significant reduction in SOD2 immunodensity was detected ($p < 0.0001$) (Figure 7E). A significant enhancement in 8-OHdG immunopositivity was found in both species from the Tusciano River (European eel: $p < 0.0001$, brown trout: $p < 0.05$) (Figure 8D).

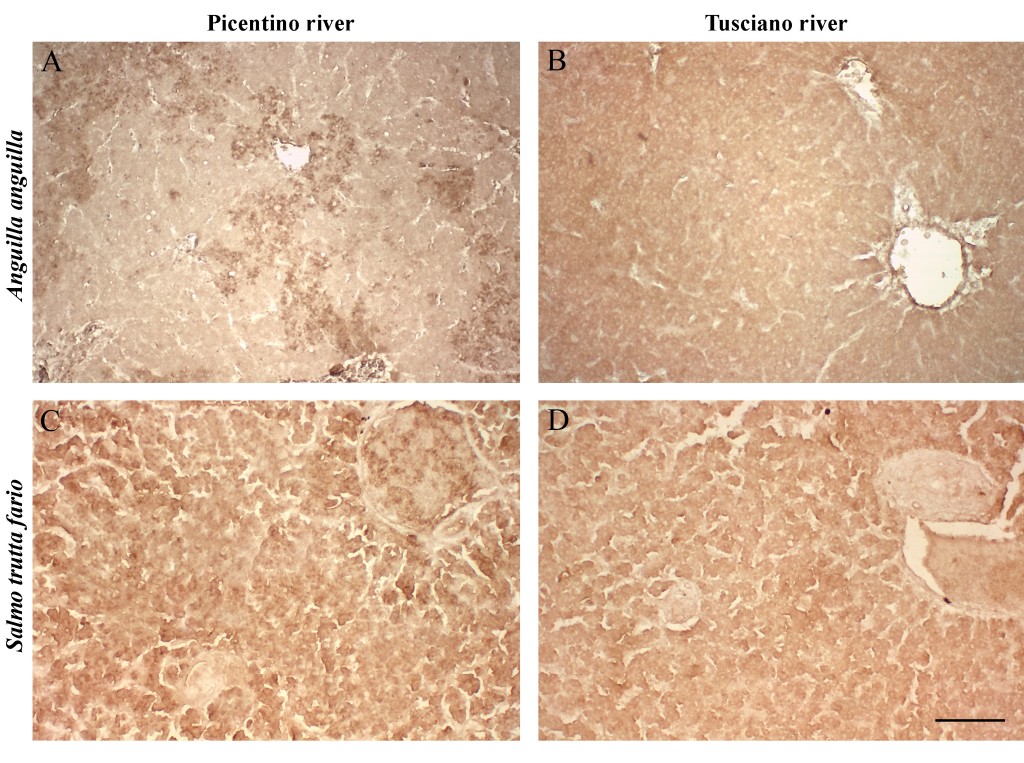

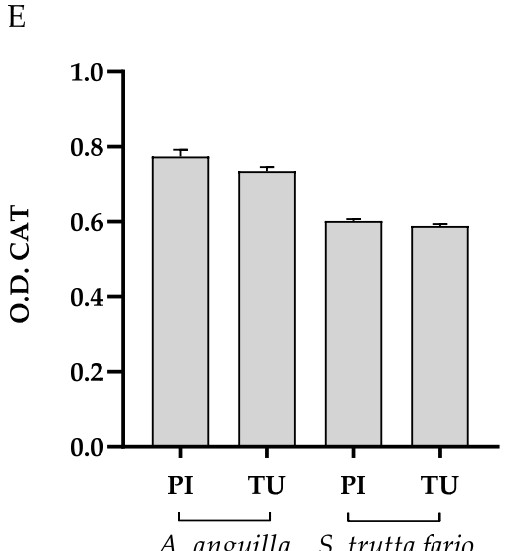

**Figure 6.** Immunohistochemical analysis of Catalase (CAT, 20×) in the liver of European eel (*A. anguilla*) from the Picentino (**A**) and Tusciano Rivers (**B**) and of brown trout (*S. trutta fario*) from the Picentino (**C**) and Tusciano Rivers (**D**). Scale bar: 150 μm. (**E**) Bar graph shows CAT optical density (O. D.) in the liver of European eel (*A. anguilla*) and brown trout (*S. trutta fario*) from the Picentino (PI) and Tusciano (TU) Rivers. Values are presented as mean ± SD.

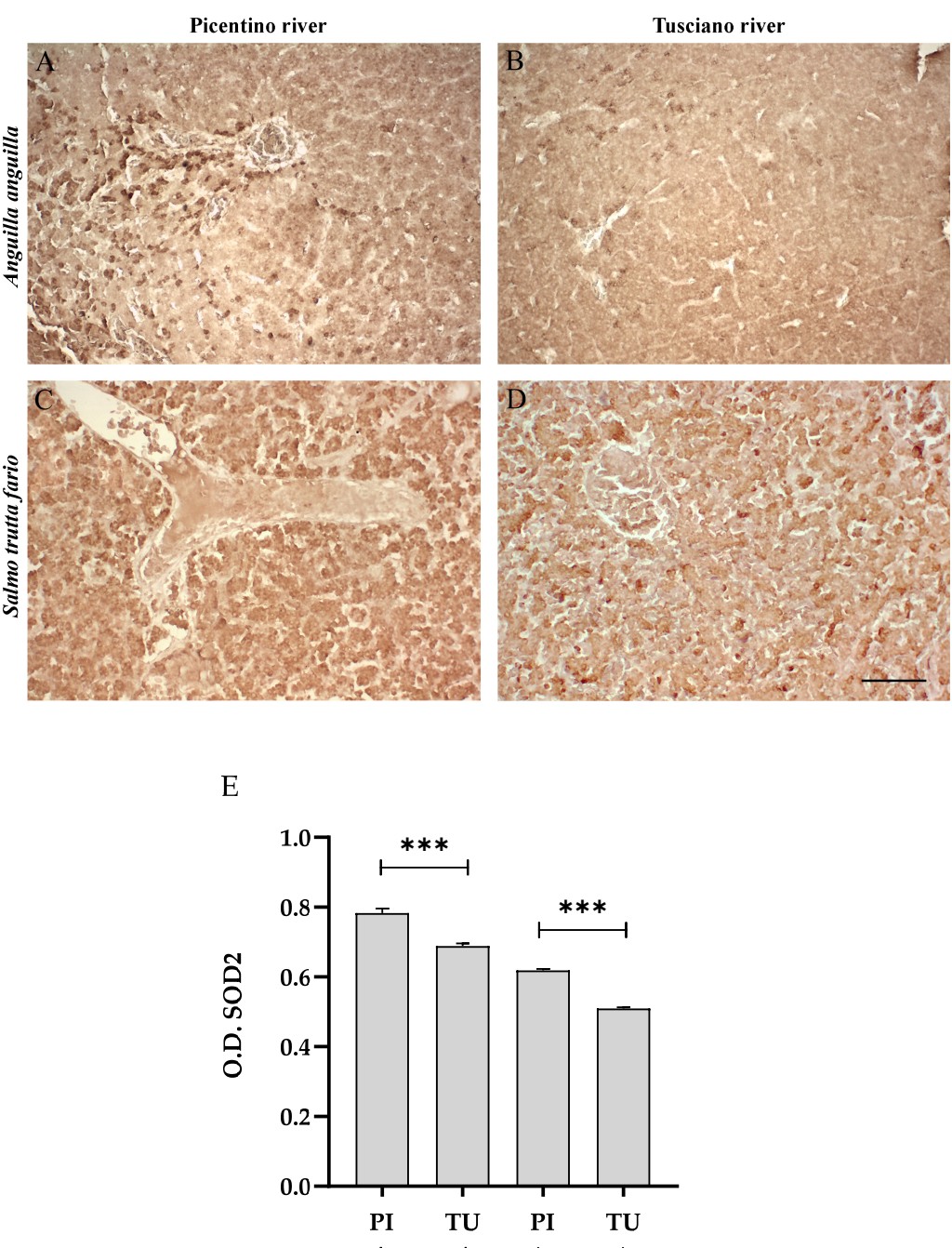

**Figure 7.** Immunohistochemical analysis of Superoxide dismutase 2 (SOD2, 20×) in the liver of European eel (*A. anguilla*) from the Picentino (**A**) and Tusciano Rivers (**B**) and of brown trout (*S. trutta fario*) from the Picentino (**C**) and Tusciano Rivers (**D**). Scale bar: 150 μm. (**E**) Bar graph shows SOD2 optical density (O. D.) in the liver of European eel (*A. anguilla*) and brown trout (*S. trutta fario*) from the Picentino (PI) and Tusciano (TU) Rivers. Values are presented as mean ± SD. * indicates statistical difference between sampled sites (*** $p < 0.0001$).

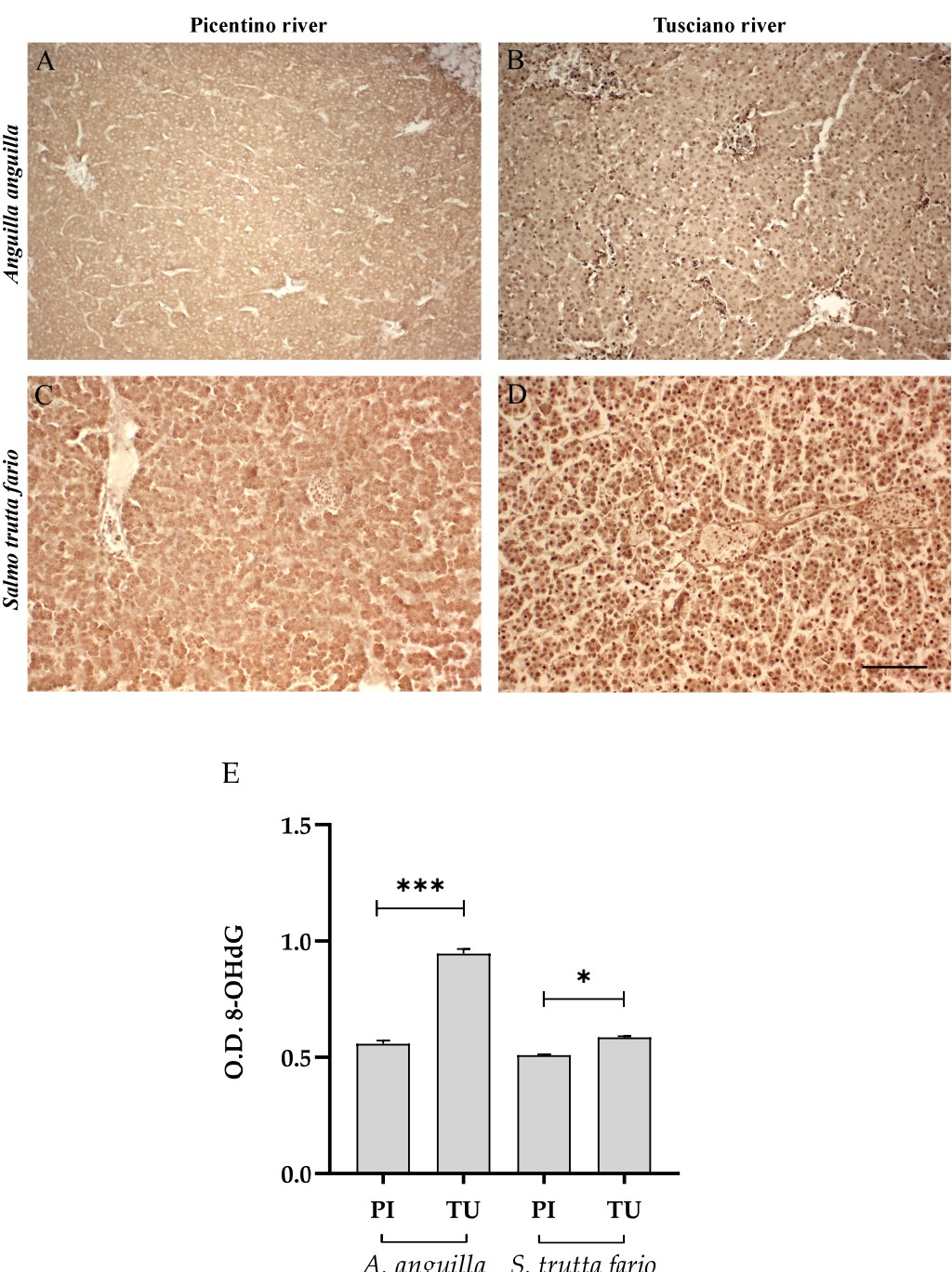

**Figure 8.** Immunohistochemical analysis of 8-Hydroxy-2′-deoxyguanosine (8-OHdG, 20×) in the liver of European eel (*A. anguilla*) from the Picentino (**A**) and Tusciano Rivers (**B**) and of brown trout (*S. trutta fario*) from the Picentino (**C**) and Tusciano Rivers (**D**). Scale bar: 150 μm. (**E**) Bar graph shows 8-OHdG optical density (O. D.) in the liver of European eel (*A. anguilla*) and brown trout (*S. trutta fario*) from the Picentino (PI) and Tusciano (TU) Rivers. Values are presented as mean ± SD. * indicates statistical difference between sampled sites (*** $p < 0.0001$; * $p < 0.05$).

### 3.5. Spleen Histopathology

The splenic sections of European eel (Figure 9A,A$_1$) and brown trout (Figure 9C,C$_1$) sampled in the Picentino River showed a normal tissue structure, normal distribution between the stromal area and the parenchymal area and almost total absence of lesions. Numerous tissue alterations were instead identified in the specimens sampled in the Tusciano River (Figure 9B,B$_1$,D,D$_1$). The main splenic alterations identified were (Table 6):

(1) MMCs; (2) hemosiderosis; (3) free melanomacrophages; and (4) necrosis. The spleen sections of fish sampled in the Tusciano River had significantly higher numbers of free melanomacrophages, aggregates of MMCs and hemosiderosis zones compared to those sampled in the Picentino River. Numerous areas of necrosis ($p < 0.0001$) were present in the spleen of the trouts sampled in the Tusciano River.

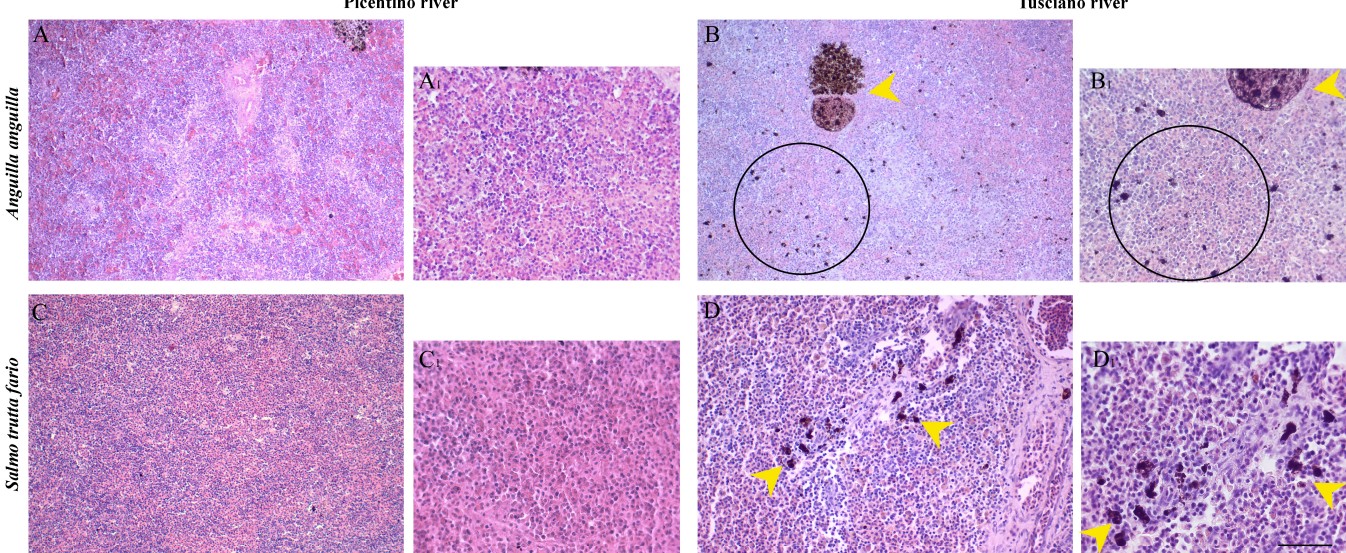

**Figure 9.** Histopathological assessment of the spleen of European eel (*A. Anguilla*) from the Picentino (**A**,**A₁**) and Tusciano Rivers (**B**,**B₁**) and in the spleen of brown trout (*S. trutta fario*) from Picentino (**C**,**C₁**) and Tusciano rivers (**D**,**D₁**) (20× and 40×, H&E). (**A**,**A₁**) Spleen of European eel from the Picentino River showing normal morphology; (**B**,**B₁**) Spleen of European eel from the Tusciano River showing accumulation of free melanomacrophages distributed in the parenchymal area (black circle), and melanomacrophages centers MMCs (yellow arrow); (**C**,**C₁**) Spleen of brown trout from the Picentino River showing normal morphology; (**D**,**D₁**) Spleen of brown trout from the Tusciano river showing accumulation of MMCs (yellow arrow). Scale bar: 150 µm, and 75 µm for the higher magnifications.

**Table 6.** Indices of histopathological alterations (HA) of spleen found in European eel (*A. Anguilla*) and brown trout (*S. trutta fario*) of the Picentino and Tusciano Rivers. Values are reported as mean ± SD. Spleen alterations were scored as follows: 0 = none, 1 = mild, 2 = moderate and 3 = severe.

| Alteration | European Eel (*A. anguilla*) | | Brown Trout (*S. trutta fario*) | |
|---|---|---|---|---|
| | Picentino River | Tusciano River | Picentino River | Tusciano River |
| Free melanomacrophages | 0.62 ± 0.52 | 2.25 ± 0.70 [c] | 0.5 ± 0.53 | 1.5 ± 0.75 [b] |
| Hemosideriosis | 0.5 ± 0.53 | 1.25 ± 0.46 [b] | 0.62 ± 0.51 | 1.37 ± 0.52 [a] |
| Melanomacrophage aggregates | 1.5 ± 0.53 | 2.5 ± 0.53 [b] | 0.37 ± 0.51 | 1.62 ± 0.52 [c] |
| Necrosis | 0.37 ± 1.36 | 1.12 ± 1.55 | 0.75 ± 1.38 | 2.62 ± 1.06 [b] |

Different letters indicate statistical difference between the samples within the river (a: $p < 0.05$; b: $p < 0.001$; c: $p < 0.0001$).

The predominant morphohistopathological lesion in the splenic samples analyzed was the alteration in the number of MMCs. Figure 10 shows the average number of MMCs per splenic section analyzed in European eel and brown trout. In both fish species sampled in the Tusciano River, a significant increase in the number of MMCs was found ($p < 0.0001$) compared to the animals collected in the Picentino River.

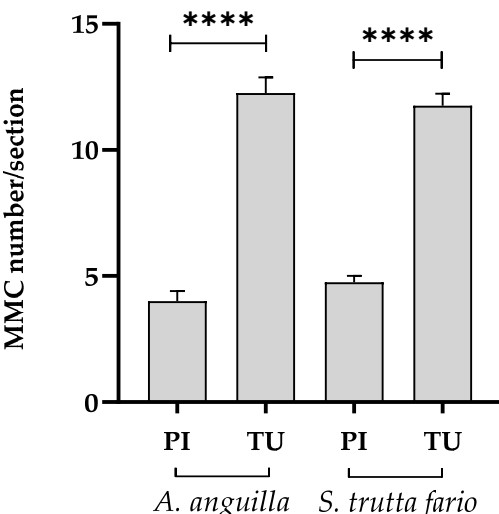

**Figure 10.** Number of melanomacrophages centers (MMCs) per splenic section in European eel (*A. anguilla*) and brown trout (*S. trutta fario*) of the Picentino (PI) and Tusciano (TU) Rivers. Values are presented as mean $\pm$ SD. * indicates statistical difference between sampled sites (**** $p < 0.0001$).

Gomori staining revealed a significant increase in the area of fibrosis in the spleen of animals from the Tusciano River (Figure 11B,D). Histological analysis revealed an increase in interstitial stroma and a decrease in parenchyma in the spleen of Tusciano River fish compared to specimens from the Picentino River (Figure 11A,C). Specifically, the percentage of the area of fibrosis exceeds 50% (51.75 $\pm$ 7.32) in the European eel and reaches 35% (35.12 $\pm$ 6.64) in the brown trout collected in the Tusciano River, showing a significant increase in areas of fibrosis compared with the species from the Picentino River (Figure 11E).

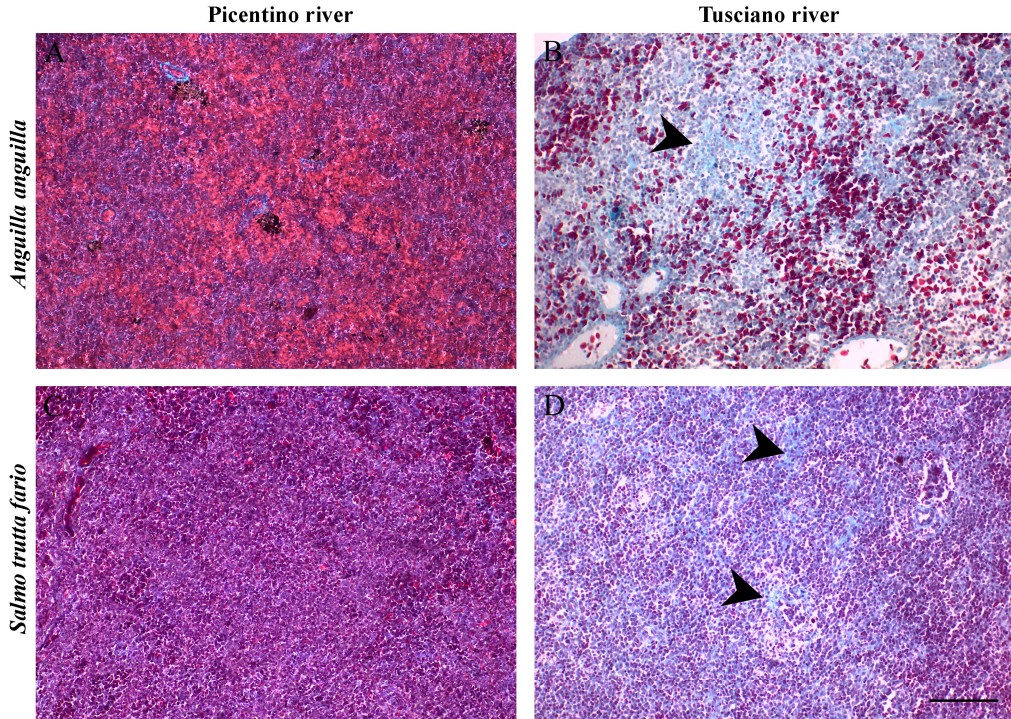

**Figure 11.** *Cont.*

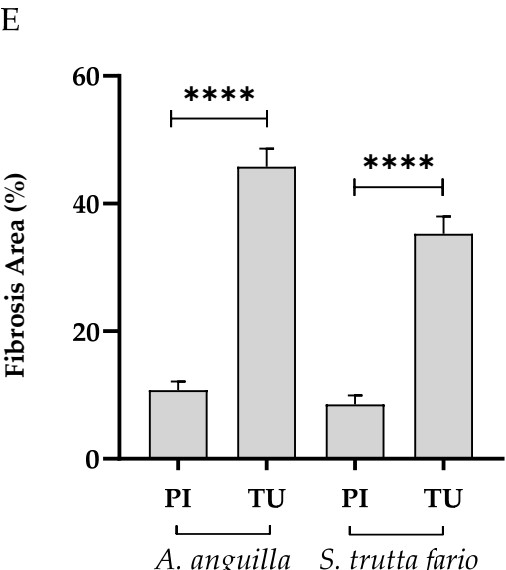

**Figure 11.** Histopathological evaluation of the spleen of European eel (*A. anguilla*) from the Picentino (**A**) and Tusciano rivers (**B**) and of the spleen of brown trout (*S. trutta fario*) from the Picentino (**C**) and Tusciano rivers (**D**). (**A,C**) splenic tissue with normal structure of fish from the Picentino River (20×; Gomori trichrome); (**B,D**) excessive production of connective tissue called fibrosis (black arrow) of fish of the Tusciano River (20×; Gomori trichrome). Scale bar: 150 μm. (**E**) quantification of fibrosis area in European eel and brown trout spleen from the Picentino and Tusciano Rivers. Values are presented as mean ± SD. * indicates statistical difference between sampled sites (**** $p < 0.0001$).

### 3.6. Spleen Immunohistochemistry

#### 3.6.1. Inflammatory Biomarkers

Figures 12 and 13 illustrate the immunopositivity of TNFα and COX2 in the spleen of European eel and brown trout from the Picentino and Tusciano Rivers. In the splenic sections of European eel, the immunopositivity of TNFα was weak or absent in the specimens in both rivers. On the other hand, in the brown trout from Picentino, a slight expression in TNFα was found, which significantly increased in the splenic cells of the specimens from the Tusciano River ($p < 0.0001$) (Figure 12). Regarding COX2 immunoexpression, in the splenic sections of European eel and brown trout from the Picentino River, a slight immunopositivity was detected, while a significant increase in COX2 immunodensity was found in both species from the Tusciano River (European eel: $p < 0.0001$, brown trout: $p < 0.05$) (Figure 13).

#### 3.6.2. Antioxidant Biomarkers

The splenic sections of European eel and brown trout sampled in the Picentino River showed immunopositivity for the antioxidant enzymes CAT and SOD2 (Figures 14A and 15A) but no immunopositivity for 8-OHdG (Figure 16A). In the spleen of both species from Tusciano River, there was a significant reduction of CAT (European eel: $p < 0.0001$, brown trout: $p < 0.0001$) and SOD2 (European eel: $p < 0.05$, brown trout: $p < 0.0001$) immunodensity (Figures 14 and 15), and a significant increase in 8-OHdG immunodensity (European eel: $p < 0.0001$, brown trout: $p < 0.0001$) (Figure 16).

### 3.7. Correlation Analysis

The following liver histopathological alterations were positively correlated with all environmental pollutants of stream water and soil: hemorrhage ($p \leq 0.05$), cytoplasmic vacuolization ($p \leq 0.01$), hemosiderosis ($p \leq 0.05$), irregular arrangement of hepatocytes ($p \leq 0.01$), lipid accumulation ($p \leq 0.05$), necrosis ($p \leq 0.01$), cellular hyperplasia ($p \leq 0.05$), leukocyte infiltration ($p \leq 0.01$) and MMC ($p \leq 0.01$). There was no correlation

between blood sinusoid dilation and water and soil pollutants (Table 7). In the spleen, only hemosiderosis correlated with water and soil pollutants ($p \leq 0.05$) (Table 8).

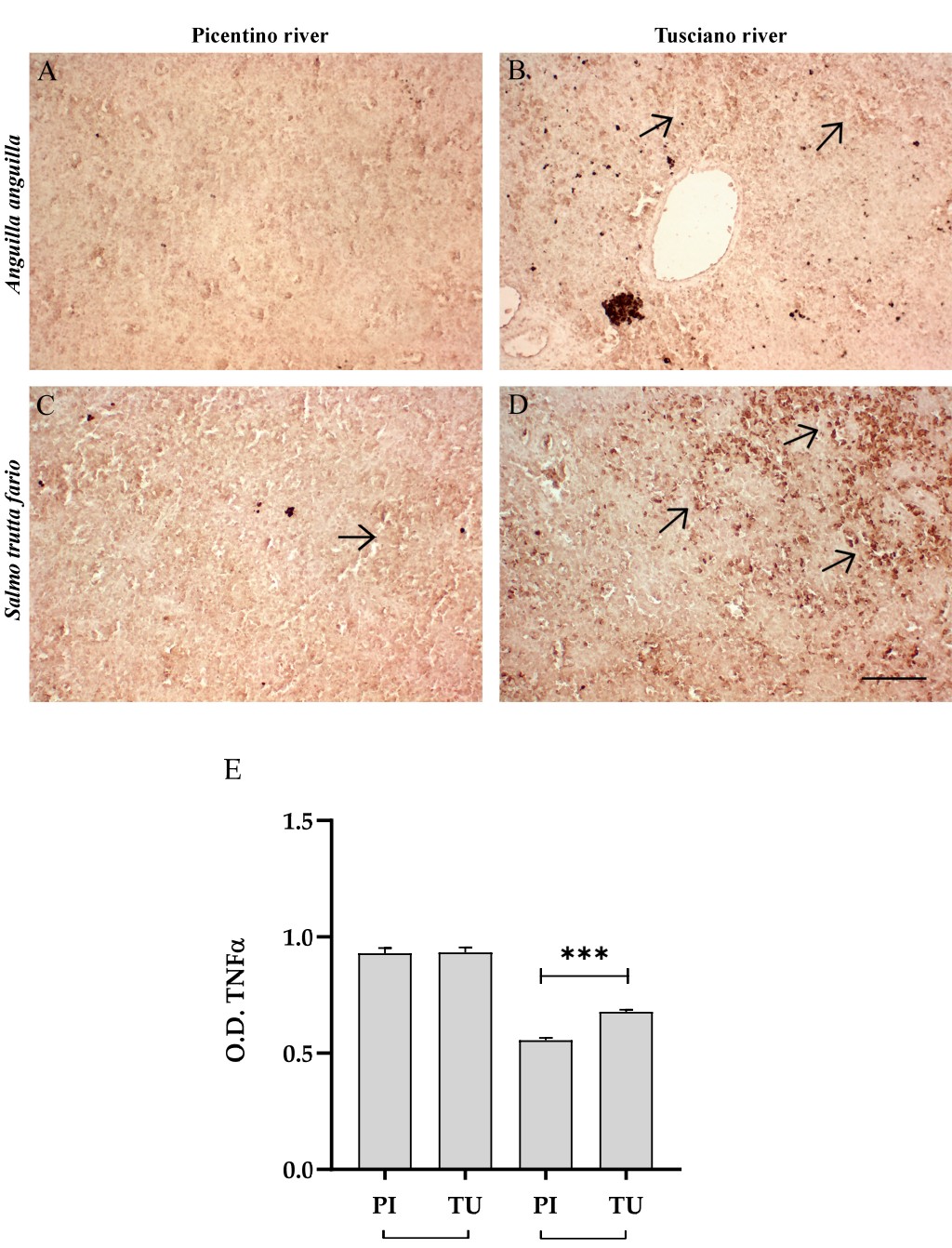

**Figure 12.** Immunohistochemical analysis of Tumor Necrosis Factor $\alpha$ (TNF$\alpha$, 20$\times$) in the spleen of European eel (*A. anguilla*) from the Picentino (**A**) and Tusciano Rivers (**B**) and of brown trout (*S. trutta fario*) from the Picentino (**C**) and Tusciano Rivers (**D**). Arrows indicate immunopositivity of inflammation biomarkers. Scale bar: 150 µm. (**E**) Bar graph shows TNF$\alpha$ optical density (O. D.) in the spleen of European eel (*A. anguilla*) and brown trout (*S. trutta fario*) from the Picentino (PI) and Tusciano (TU) Rivers. Values are presented as mean $\pm$ SD. * indicates statistical difference between sampled sites (*** $p < 0.0001$).

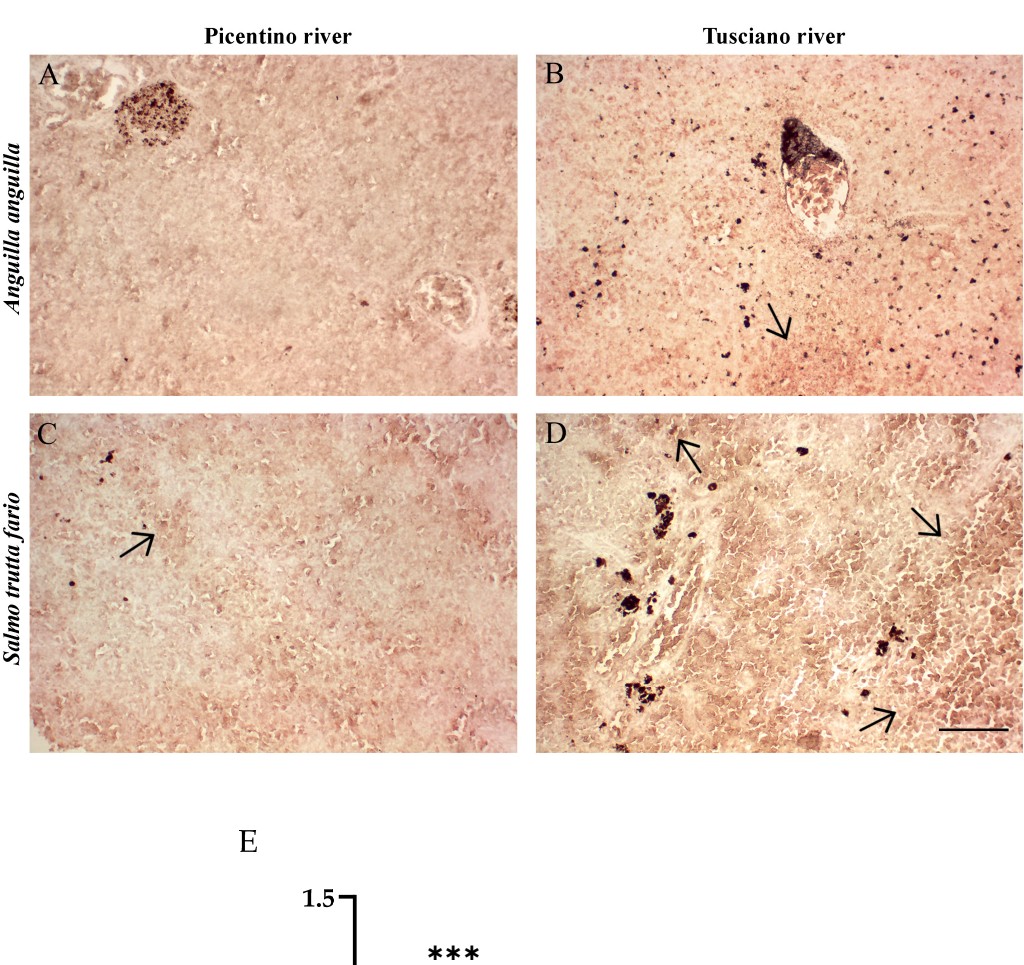

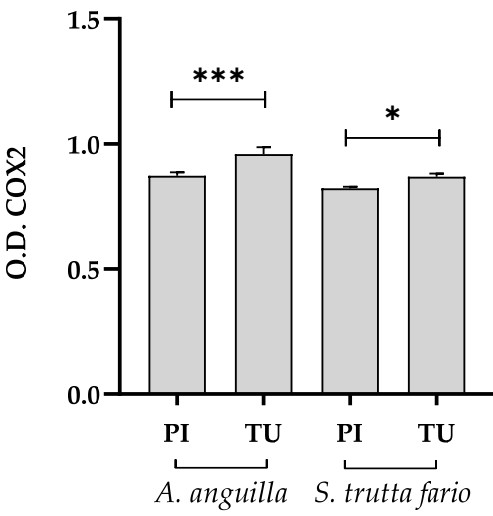

**Figure 13.** Immunohistochemical analysis of Cycloxygenase 2 (COX2, 20×) in the spleen of European eel (*A. anguilla*) from the Picentino (**A**) and Tusciano Rivers (**B**) and of brown trout (*S. trutta fario*) from the Picentino (**C**) and Tusciano Rivers (**D**). Arrows indicate immunopositivity of inflammation biomarkers. Scale bar: 150 μm. (**E**) Bar graph shows COX2 optical density (O. D.) in the spleen of European eel (*A. anguilla*) and in brown trout (*S. trutta fario*) from the Picentino (PI) and Tusciano (TU) Rivers. Values are presented as mean ± SD. * indicates statistical difference between sampled sites (*** $p < 0.0001$; * $p < 0.05$).

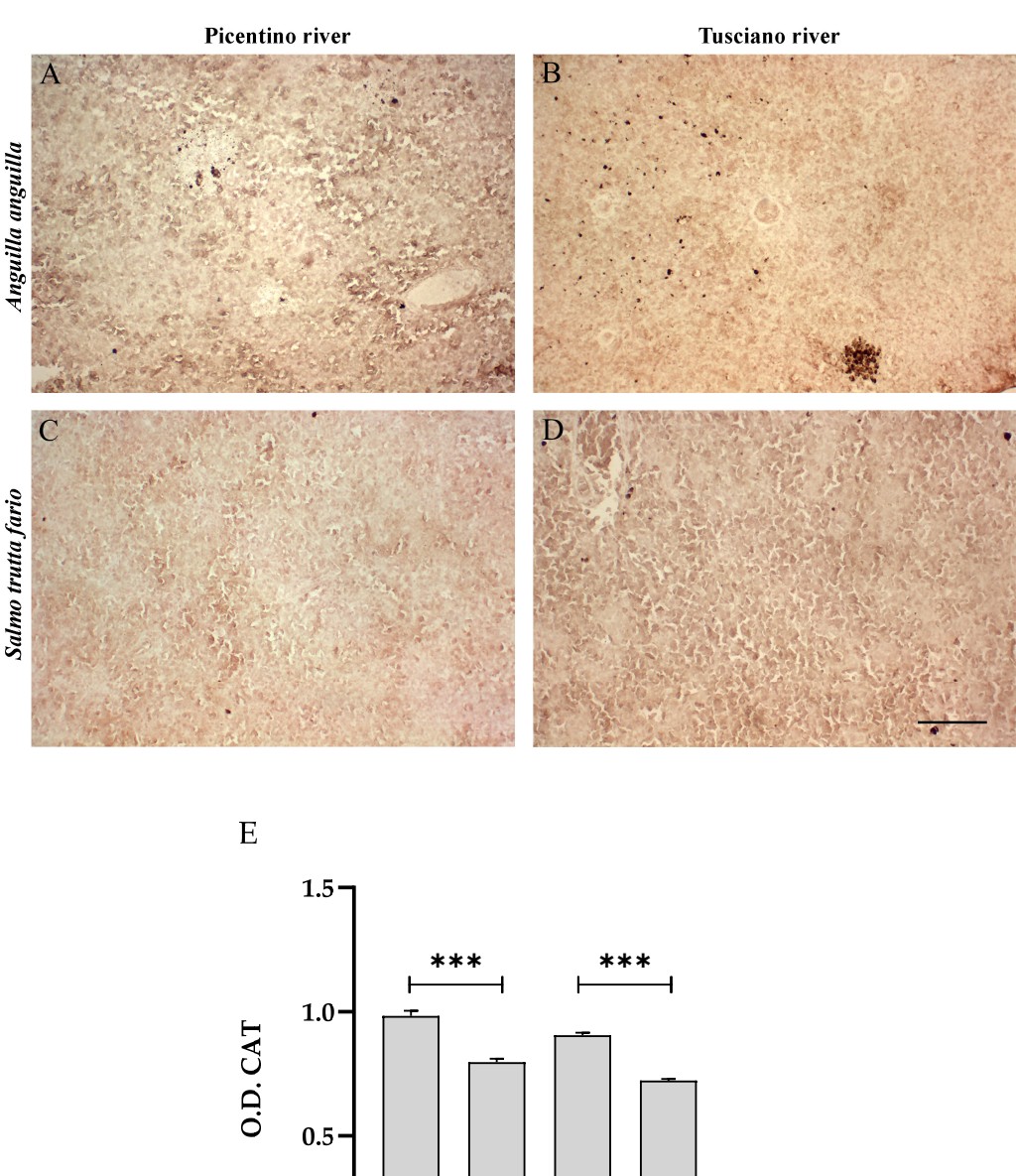

**Figure 14.** Immunohistochemical analysis of Catalase (CAT, 20×) in the spleen of European eel (*A. anguilla*) from the Picentino (**A**) and Tusciano Rivers (**B**) and of brown trout (*S. trutta fario*) from the Picentino (**C**) and Tusciano Rivers (**D**). Scale bar: 150 μm. (**E**) Bar graph shows CAT optical density (O. D.) in the spleens of European eel (*A. anguilla*) and brown trout (*S. trutta fario*) from the Picentino (PI) and Tusciano (TU) Rivers. Values are presented as mean $\pm$ SD. * indicates statistical difference between sampled sites (*** $p < 0.0001$).

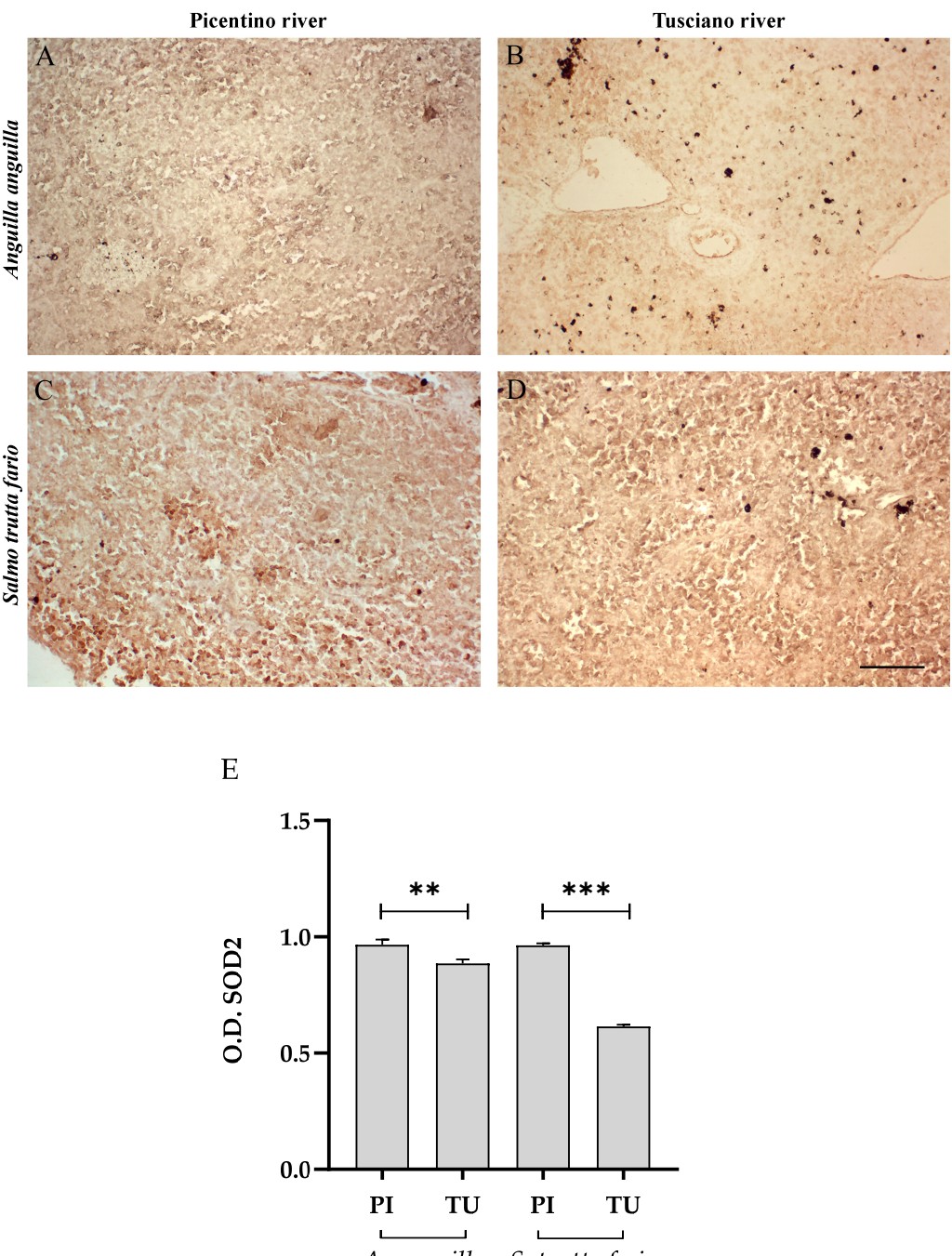

**Figure 15.** Immunohistochemical analysis of Superoxide dismutase 2 (SOD2, 20×) in the spleen of European eel (*A. anguilla*) from the Picentino (**A**) and Tusciano Rivers (**B**) and of brown trout (*S. trutta fario*) from the Picentino (**C**) and Tusciano Rivers (**D**). Scale bar: 150 μm. (**E**) Bar graph shows SOD2 optical density (O. D.) in the spleen of European eel (*A. anguilla*) and brown trout (*S. trutta fario*) from the Picentino (PI) and Tusciano (TU) Rivers. Values are presented as mean ± SD. * indicates statistical difference between sampled sites (*** $p < 0.0001$; ** $p < 0.001$).

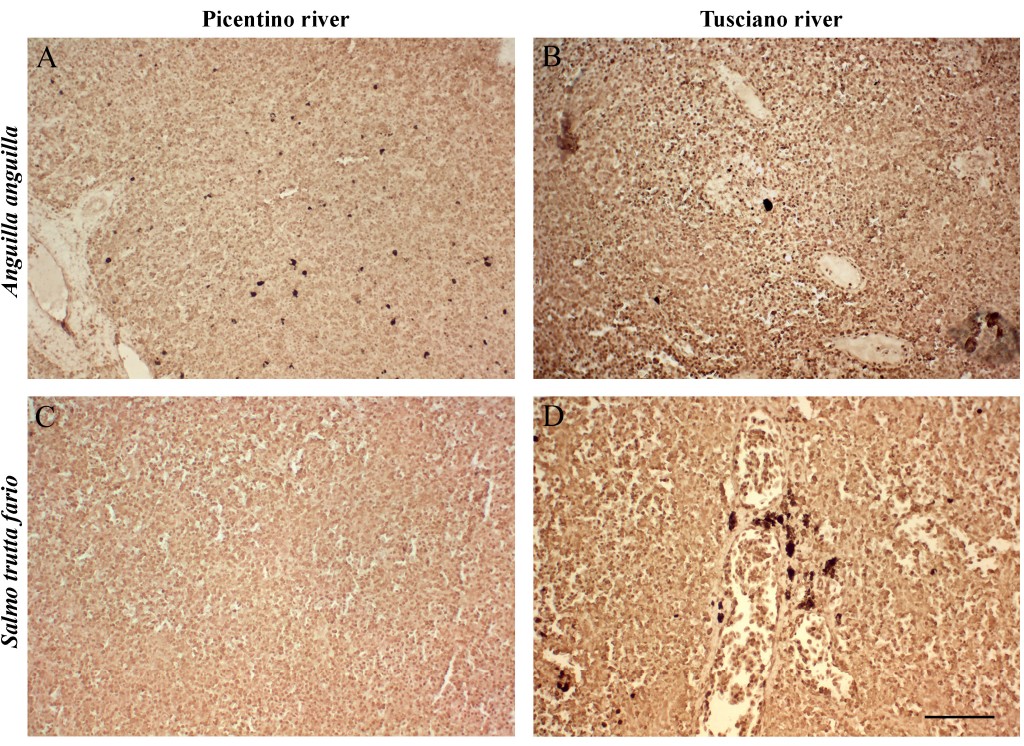

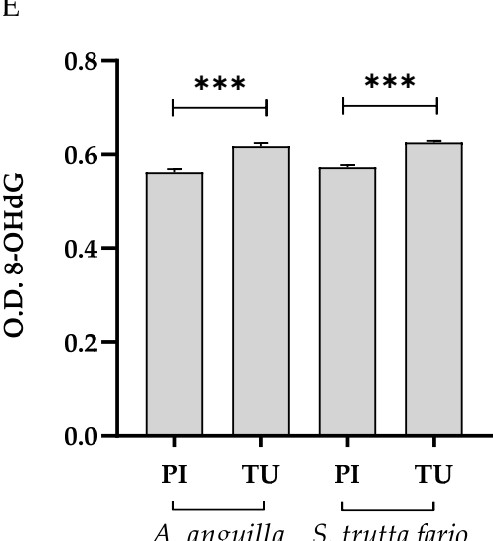

**Figure 16.** Immunohistochemical analysis of 8-Hydroxy-2′-deoxyguanosine (8-OHdG, 20×) in the spleen of European eel (*A. anguilla*) from the Picentino (**A**) and Tusciano Rivers (**B**) and of brown trout (*S. trutta fario*) from the Picentino (**C**) and Tusciano Rivers (**D**). Scale bar: 150 μm. (**E**) Bar graph shows 8-OHdG optical density (O. D.) in the spleen of European eel (*A. anguilla*) and brown trout (*S. trutta fario*) from the Picentino (PI) and Tusciano (TU) Rivers. Values are presented as mean ± SD. * indicates statistical difference between sampled sites (*** $p < 0.0001$).

**Table 7.** Correlation analysis of liver histopathological alterations and environmental pollutants in stream water and soil. *p*-value is from a two-tailed test with a confidence interval of 95%. Statistical significance is represented by stars as follows: nonsignificant (ns) > 0.05, * $p \leq 0.05$, ** $p \leq 0.01$.

| | **Water Pollutants** | | | **Soil Pollutants** | | |
|---|---|---|---|---|---|---|
| **Alterations** | **Pearson r** | **p Value (Two-Tailed)** | **Statistical Significance** | **Pearson r** | **p Value (Two-Tailed)** | **Statistical Significance** |
| Blood sinusoid dilation | 0.9417 | 0.0583 | ns | 0.9417 | 0.0583 | ns |
| Hemorrhage | 0.9886 | 0.0114 | * | 0.9886 | 0.0114 | * |
| Cytoplasmic vacuolization | 0.9939 | 0.0061 | ** | 0.9939 | 0.0061 | ** |
| Hemosiderosis | 0.9874 | 0.0126 | * | 0.9874 | 0.0126 | * |
| Irregular arrangement of hepatocytes | 0.9962 | 0.0038 | ** | 0.9962 | 0.0038 | ** |
| Lipid accumulation | 0.9611 | 0.0389 | * | 0.9611 | 0.0389 | * |
| Necrosis | 0.9921 | 0.0079 | ** | 0.9921 | 0.0079 | ** |
| Cellular hyperplasia | 0.9831 | 0.0169 | * | 0.9831 | 0.0169 | * |
| Leukocytes infiltration | 0.9923 | 0.0077 | ** | 0.9923 | 0.0077 | ** |
| MMC | 0.9925 | 0.0075 | ** | 0.9925 | 0.0075 | ** |

**Table 8.** Correlation analysis of spleen histopathological alterations and environmental pollutants in stream water and soil. *p*-value is from a two-tailed test with a confidence interval of 95%. Statistical significance is represented by stars as follows: nonsignificant (ns) > 0.05, * $p \leq 0.05$.

| | **Water Pollutants** | | | **Soil Pollutants** | | |
|---|---|---|---|---|---|---|
| **Alterations** | **Pearson r** | **p Value (Two-Tailed)** | **Statistical Significance** | **Pearson r** | **p Value (Two-Tailed)** | **Statistical Significance** |
| Free melanomacrophages | 0.9258 | 0.0742 | ns | 0.9258 | 0.0742 | ns |
| Hemosiderosis | 0.9874 | 0.0126 | * | 0.9874 | 0.0126 | * |
| Melanomacrophage aggregates | 0.7432 | 0.2568 | ns | 0.7432 | 0.2568 | ns |
| Necrosis | 0.7675 | 0.2325 | ns | 0.7675 | 0.2325 | ns |

*3.8. Erythrocyte Abnormalities*

Different ENA, such as micronucleus, blebbed and notched nuclei and ECA, such as deformations and elongations, were observed in the blood of European eel and brown trout from the Tusciano River compared with the Picentino River (Table 9). Specifically, a significant increase in ENA was found in the blood of both European eel and brown trout from the Tusciano River compared with the Picentino River ($p < 0.001$). Similar results were obtained for ECA with a significant increase of deformed and elongated erythrocytes in the European eel ($p < 0.0001$) and brown trout ($p < 0.001$) from the Tusciano River compared with the Picentino River. On the contrary, EF showed an increase only in the brown trout of the Tusciano River compared with the Picentino River ($p < 0.001$).

**Table 9.** Percentage of erythrocytic cellular abnormalities (ECA), erythrocytic fusion (EF) and erythrocytic nuclear abnormalities (ENA) in European eel (*A. anguilla*) and brown trout (*S. trutta fario*) from the Picentino and Tusciano Rivers.

| | | **ECA** | **EF** | **ENA** |
|---|---|---|---|---|
| European eel (*A. anguilla*) | Picentino River | 12.7 ± 2 | 1.18 ± 0.3 | 7.7 ± 2.1 |
| | Tusciano River | 27.15 ± 2 *** | 2.8 ± 0.3 | 20.3 ± 0.9 ** |
| Brown trout (*S. trutta fario*) | Picentino River | 16.3 ± 0.9 | 16.3 ± 0.9 | 9.1 ± 1.6 |
| | Tusciano River | 28.6 ± 1.9 ** | 28.6 ± 1.9 *** | 21.1 ± 2.8 ** |

*** $p < 0.0001$ and ** $p < 0.001$. All values expressed as mean ± SE. Three slides were prepared from each fish, 1000 cells were scored from each slide, and at least three fishes were analyzed from each group.

## 4. Discussion

In this paper, we report the assessment of the health status of two of the most abundant and widely distributed fish species: the European eel (*Anguilla anguilla*) and the brown trout (*Salmo trutta fario*). As reported by scientific studies, fish represent an excellent tool in

the biomonitoring of aquatic environments [4,34]. However, as suggested by Griffiths [35], the use of only one fish species as a bioindicator of an aquatic environment may cause some problems mainly related to the narrowness of the fish zoning. For this reason, we selected two fish species widely distributed in the Mediterranean rivers and commonly employed in the assessment of the impacts of anthropogenic activities [36,37].

The analysis of biological indices provided interesting information about the health status of the species living in the Picentino and Tusciano Rivers. The analysis of the condition index K, a noninvasive indicator of fish health [38], showed values similar to the reference scores for European eel [39] and brown trout [40], suggesting a good nutritional status in both river locations. On the contrary, the hepatosomatic index (HSI), another useful indicator of the physiological state of fish [18], was significantly higher in both species living in the Tusciano River. Numerous scientific studies report that exposure to aquatic pollutants (including heavy metals) can lead to an increase in the HSI of fish [17,41,42] due to the increment in detoxification activity and lipid accumulation in the liver [43]. The high HSI value found in the European eel and brown trout living in the Tusciano River is, therefore, an indication of pollutant exposure.

Among the ecotoxicological methods, histopathology is a useful tool for detecting the toxic effects of pollutants, allowing the localization and quantification of lesions induced by contaminants on key organs, such as the spleen and liver [12,44]. In fish, the liver carries out vital functions, from basic metabolism to accumulation, biotransformation and excretion of contaminants [2,21]; thus, liver tissue changes are indicative of fish health status [44]. Morphohistopathological analysis has shown a significant increase in the number of liver alterations in the European eel and brown trout from the Tusciano River compared to the Picentino River. Numerous liver histopathological lesions were observed, including degeneration of the liver parenchyma, deformation of hepatocytes, cytoplasmic vacuolization and dilation of the hepatic capillaries. It has been reported that cytoplasmatic vacuolization represents the most pathological lesion of fish exposed to aquatic contaminants [45]. Indeed, exposure to heavy metals induces metabolic damage, leading to a degenerative process that causes the accumulation of fat in the form of vacuoles in the cytoplasm of the liver [2,46]. Several reports have demonstrated that there is a correlation between exposure to contaminants and the development of toxicopathic and histocytopathological hepatic lesions in fish [47–49]. Other important histopathological lesions found in the liver of the European eel and brown trout in the Tusciano River that could represent the consequence of exposure to environmental contaminants are hemorrhage, congestion in sinusoids, and leukocyte infiltration. As reported by Javed et al. [50], hepatic hemorrhage may result from the congestion of blood vessels due to the increase in blood pressure induced by exposure to toxic substances. Similarly, leukocyte infiltration, one of the main alterations typical of the inflammatory process, is another morphological disorder commonly found in the liver of organisms exposed to pollutants. Kaur et al. [51] reported the presence of hepatic cellular infiltrations as an index of activation of the immune system following exposure to environmental contaminants, and, according to Bernet et al. [12], leukocyte infiltration can be classified as a tissue alteration of moderate severity.

Using the factors of importance ($\omega$) and the score values (a) of all the alterations found in the liver, the histopathological index of the liver of both fish species from the Tusciano River showed a significant increase when compared with the Picentino River. As reported by Bernet et al. [12], the histopathological index represents the degree of damage of an organ, and a high histopathological index indicates a high degree of organ damage. Therefore, the increase in the histopathological index of the liver identified in the European eel and brown trout from the Tusciano River suggests the presence of mild/moderate liver damage compared to the Picentino River.

Altered tissue morphology was observed in the spleen of the European eel and brown trout sampled in the Tusciano River. The main splenic histopathological changes included the presence of free melanomacrophages and hemosiderosis, the increase in connective tissue (fibrosis) and the increase in the number of MMCs. The MMCs of the fish spleen

represent one of the most important physiological characteristics to be evaluated in the biomonitoring studies of aquatic species. It has been reported that stressful conditions result in an increase in the number and size of splenic MMCs [18]. In the spleen, MMCs are mainly located near the blood vessels and are an integral part of the immune system. In the presence of environmental pollution, an increase in MMCs is expected due to robust phagocytic activity [52]. Our results show an increase in the number of MMCs in the splenic sections of both fish species of the Tusciano River compared to the Picentino River. These results, in agreement with the studies conducted by Araújo et al. [53] and Passantino et al. [54], suggest that the increase in splenic MMCs in fish from the Tusciano River may be linked to exposure to aquatic contaminants.

Both European eel and brown trout from the Tusciano River showed a high presence of connective tissue (a condition called fibrosis) in the spleen. Similar levels of fibrosis have been documented in fish exposed to different pollutants, including pesticides [55,56], heavy metals [18,57] and contaminating waste [58]. Studies conducted by Vieira et al. [59] and Nai et al. [60] report that exposure to high concentrations of heavy metals (such as lead, mercury, chromium and nickel) alters ion homeostasis and induces DNA damage, causing various diseases, including splenic fibrosis. Large areas of fibrosis, therefore, suggest a higher exposure to pollutants in fish from the Tusciano River compared to Picentino River.

Since the liver and spleen histopathological investigations suggested the presence of morphological damage, we decided to carry out an immunohistochemical analysis to monitor the inflammation and the antioxidant defense status. The hepatic and splenic immunohistochemical results suggest the presence of an inflammatory state in both the European eel and brown trout living in the Tusciano River. TNFα is an important pro-inflammatory cytokine involved in several activities, such as cell proliferation, inflammation and apoptosis [61]. An increasing number of scientific research reports that exposure to environmental contaminants affects the expression and function of TNFα [1,23]. Similarly, the enzyme COX2, being induced by inflammatory stimuli such as bacterial endotoxins and cytokines, is closely linked to fish's innate immune response [62], and its expression increases in the presence of tissue damage [63].

Oxidative stress occurs when there is a physiological imbalance in favor of oxidants between the levels of antioxidants and oxidants (free radicals or oxygen-reactive species). Among the antioxidant defense system, there are the enzymes SOD2 and CAT. These enzymes are considered the vital first-line defense against oxidative stress. The hepatic and splenic immunohistochemical results suggest an impairment of the antioxidant defense system in both the European eel and brown trout living in the Tusciano River, characterized by an almost total absence of CAT immunoexpression and a poor SOD2 immunoexpression, accompanied by an increase in 8-OHdG immunoexpression, the main biomarker of DNA damage induced by oxidative stress. Studies suggest that exposure to aquatic contaminants and/or stressors induces alterations in the fish antioxidant system [5,64]. Environmental pollutants cause an increase in reactive oxygen species (ROS) and disturb the efficiency of the cellular antioxidant enzymatic system. When the physiological conditions are optimal, the antioxidant defense system is able to reduce the ROS induced by the exposure to exogenous compounds. On the contrary, when excessive production of ROS occurs, an imbalance of the antioxidant defense system takes place, resulting in oxidative damage and suppression of the antioxidant activity of SOD2 and CAT [5]. The reduction in SOD2 and CAT activity observed in the European eel and brown trout from the Tusciano River may reflect an altered antioxidant defense system or a direct effect of toxic substances altering the activity of such enzymes [64,65]. The results of the 8-hydroxy-2-deoxyguanosine (8-OHdG) immunohistochemical analysis also confirm the hypothesis of a contamination of the aquatic environment of the Tusciano River. 8-OHdG is one of the most studied biomarkers of DNA oxidative damage induced by ROS generated from normal metabolisms and several environmental factors. The immunoexpression of the oxidative stress marker 8-OHdG was, in fact, elevated in the splenic and hepatic tissues of the specimens collected in the Tusciano

River, suggesting a high degree of DNA oxidative damage induced by the excessive production of ROS, likely caused by exposure to toxic agents and/or pollutants [66].

Another important indicator of fishes' physiological conditions is represented by blood parameters, relevant markers of environmental changes and stress in fish [67,68]. Specifically, some cellular and nuclear morphological alterations of erythrocytes can be used as biomarkers to assess the stress caused by environmental pollutants [33,69,70]. In line with this, we found an increase in the cellular and nuclear morphology alterations in both species living in the Tusciano River.

The results obtained in the present study show that there is certainly a correlation between the histopathological alterations of fish tissues and the quality of water and soil since the higher presence of pollutants is reflected in the morphological alterations of the tissues. The liver was the most sensitive organ, while the spleen was the least responsive. Although it is not possible to attribute the observed alterations to a specific water or soil parameter, the present study demonstrates that histopathology represents a useful tool for aquatic ecosystem biomonitoring and paves the way for the possibility of using liver and some of its alterations as possible indicators. Even more interesting is the possibility of using a noninvasive blood sample in biomonitoring programs to evaluate environmental stress.

## 5. Conclusions

In conclusion, our results show clear differences in the fish health status related to different water quality. Differences in histopathological lesions, inflammation and antioxidant defense systems were observed in the liver and spleen of the European eel and brown trout living in the Picentino and Tusciano Rivers. The histological and immunohistochemical multibiomarker approach used in this study could be used to evaluate the environmental safety of aquatic ecosystems around the world.

**Author Contributions:** Conceptualization, M.P.; methodology, M.P.; software, D.C., E.C. and G.O.; validation, G.O., D.C. and E.C.; formal analysis, G.O. and R.I.; investigation, G.O., R.I. and G.R.; resources, M.P.; data curation, M.P.; writing—original draft preparation, G.O. and R.I.; writing—review and editing, M.P. and G.O.; supervision, M.P.; project administration, G.O.; funding acquisition, M.P. All authors have read and agreed to the published version of this manuscript.

**Funding:** This research was funded by Campania Region as part of the Project "Innovazione, sviluppo e sostenibilità nel settore della pesca e dell'acquacoltura per la Regione Campania (ISSPA)"—PO FEAMP 2014-2020 Misura 1.44 to Marina Paolucci.

**Data Availability Statement:** The raw data supporting the conclusions of this article will be made available by the authors without undue reservation.

**Acknowledgments:** The authors gratefully acknowledge the data support from the Campania Region by means of the framework of the "Campania Trasparente—Attività di monitoraggio integrato per la Regione Campania".

**Conflicts of Interest:** The authors declare no conflict of interest.

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
