# Peer review of "A Deep Survey of Fish Health for the Recognition of Useful Biomarkers to Monitor Water Pollution"

_environments, doi:10.3390/environments10120219_

Round 1

Reviewer 1 Report

Comments and Suggestions for Authors

The paper adresses the biomonitoring of two rivers of the Campania region, located in the South of Italy.

The authors provide a research work made with the best research standard. Pages 8 to 22 contain a large core of meticulous results.The principal questions that may be aded are precisions on :

  • Age, estimation of individuals, weight or others
  • Sampling season, winter or others 
  • Non standard period avoidance, Covid, flood...
  • Blue Green transition, between sea and river, eventually

The principal interrogation in the paper is page 23 tables 7 and 8, where water pollutants and soil pollutants are explicitly presented. It is not clear where is the distinction justification in the study. It could be precised, urban, agriculture, industry, line 91.

Here follows some remarks :

-Metals line71, line 636

-PTEs choice, table3 and line 638 could be linked.

-Dot line 578

-Vie milieu is Vie et Milieu/Life & Environment line 760

Author Response

Manuscript ID: environments-2729800

The authors would like to thank the reviewers for theirs thorough reviews and suggestions. They helped improve the quality of the manuscript.

 Reviewers' comments:

 Reviewer #1:

The paper addresses the biomonitoring of two rivers of the Campania region, located in the South of Italy.

The authors provide a research work made with the best research standard. Pages 8 to 22 contain a large core of meticulous results.

Authors' reply:

We thank the reviewer

-The principal questions that may be added are precisions on:

Age, estimation of individuals, weight or others. Sampling season, winter or others. Non-standard period avoidance, Covid, flood...

Authors' reply:

Age and sampling season have been added to the revised manuscript. In particular, the age was evaluated as follows:

The age assessment of the sampled trout was carried out on the basis of the relationship (reported in the table) between age and length and body weight (Sivka et al., 2012; Verreycken et al., 2011; Badino et al., 1994; Svetovidov , 1984).

Age

weight(g)

lenght (cm)

juveniles

2 -24

4.8 – 12.5

Adults

>25

>13

The age assessment of the sampled eels was carried out on the basis of the relationship (reported in the table) between age and length and body weight (Matić-Skoko et al., 2012; Verreycken et al., 2011; Dekker et al., 1998 ).

Age

Weight (g)

Lenght (cm)

Juveniles

2 - 29

6.8 -28

Adults

>30

>29

Badino G., Lodi E., Malacarne G., Maiorana G., 1994.Tattiche riproduttive in Salmo trutta L.(Osteichthyes, Salmonidae). Atti V Convegno Naz. A.I.I.A.D. Vicenza: 37-44.

Dekker, W., B. van Os and J. van Willigen, 1998. Minimal and maximal size of eel. L'ANGUILLE EUROPEENNE. 10E REUNION DU GROUPE DE TRAVAIL "ANGUILLE" EIFAC/ICES.Bulletin Francais de Peche et Pecherie, Conseil superieur de la peche, Paris (France), 1998.

Matić-Skoko S, Josipa Ferri, Pero Tutman, Daria Skaramuca, Domagoj Đikić, Duje Lisičić, ZdenkoFranić & Boško Skaramuca (2012): The age, growth and feeding habits of the European conger eel, Conger conger (L.) inthe Adriatic Sea, Marine Biology Research, 8:10, 1012-1018

Sivka U, Halačka K, Sušnik Bajec, 2012. Morphological differences in the skin of marble trout Salmo marmoratus and of brown trout Salmo trutta. Folia Histochem Cytobiol. 2012 Jul 4;50(2):255-62. doi: 10.5603/fhc.2012.0034. PMID: 22763959.

Svetovidov, A.N., 1984. Salmonidae. p. 373-385. In P.J.P. Whitehead, M.-L. Bauchot, J.-C. Hureau, J. Nielsen and E. Tortonese (eds.) Fishes of the north-eastern Atlantic and the Mediterranean. UNESCO, Paris. vol. 1

Verreycken, H., G. Van Thuyne and C. Belpaire, 2011. Length-weight relationships of 40 freshwater fish species from two decades of monitoring in Flanders (Belgium). J. Appl. Ichthyol. 2011:1-5.

Weights are reported in lines 112-113 (n. 10, 5/site) and in Table 4 respectively.

The sampling was carried out during the covid-19 period, in compliance with the rules.

-Blue Green transition, between sea and river, eventually

Authors' reply:

In this work we do not have the parameters useful for calculating blue and green water flows. They certainly could give additional information on the water quality of the two rivers, but it is outside the scope of this work.

-It could be precised, urban, agriculture, industry, line 91.

Authors' reply:

We thank the reviewer for pointing out this shortcoming. We have inserted some lines of text (from line 91).

-The principal interrogation in the paper is page 23 tables 7 and 8, where water pollutants and soil pollutants are explicitly presented. It is not clear where is the distinction justification in the study.

Authors' reply:

The use of PTE's and POP's concentration data in both soils and waters of the studied sites is sustained by the evidence that the concentration of pollutants in both environmental matrices provides a more complete picture of the environmental health status of the investigated sites.

Organic pollutants and heavy metals present in soils usually migrate in both vertical (soil erosion, animal carrying) and horizontal (wind, surface runoff) directions. Soil is not only a site for sedimentation of pollutants, but also an important source of these pollutants to the water environment.

Here follows some remarks :

-Metals line71, line 636  

Authors' reply:

done

-PTEs choice, table 3 and line 638 could be linked.

Authors' reply:

We clarified the relationship in the period at the lines 646-649

-Dot line 578

Authors' reply:

done

-Vie milieu is Vie et Milieu/Life & Environment line 760

Authors' reply:

done

Reviewer 2 Report

Comments and Suggestions for Authors

The paper by Orso et al. deals with the use of some fish species to monitor water pollution. The paper is well written and understandable, and suitable to be published after a minor revision.

 Line 50: I think you should put a "that" between “habitat” and “make.

Line 127: I think these are the magnifications of the objectives, not the total ones.

Figure 2: probably add some histological image with major magnification would make it easier to highlight the damage described. The same for Figure 9.

Line 565: better replace “work” with “paper”.

Author Response

Manuscript ID: environments-2729800

 The authors would like to thank the reviewers for theirs thorough reviews and suggestions. They helped improve the quality of the manuscript.

 Reviewers' comments:

Reviewer #2:

The paper by Orso et al. deals with the use of some fish species to monitor water pollution. The paper is well written and understandable, and suitable to be published after a minor revision.

Authors' reply:

We thank the reviewer

 Line 50: I think you should put a "that" between “habitat” and “make.

Authors' reply:

Done

Line 127: I think these are the magnifications of the objectives, not the total ones.

Authors' reply:

Correct we clarified it in the revised manuscript

Figure 2: probably add some histological image with major magnification would make it easier to highlight the damage described. The same for Figure 9.

Authors' reply:

Magnifications have been added

Line 565: better replace “work” with “paper”.

Authors' reply:

done